# Analysis on the Higher Education Sustainability in China Based on the Comparison between Universities in China and America

**Yong-Ming He [1,2,\*]** **, Yu-Long Pei [1,3,\*], Bin Ran [2,4], Jia Kang [1,3] and Yu-Ting Song [1,3]**

1   School of Transportation, Northeast Forestry University, 26 Hexing Road, Harbin 150040, China; kangjia@nefu.edu.cn (J.K.); songyuting@nefu.edu.cn (Y.-T.S.)

2   Department of Civil and Environmental Engineering, University of Wisconsin-Madison, Madison, WI 57305, USA; yhe275@wisc.edu

3   Transport Research Centre, Northeast Forestry University, 26 Hexing Road, Harbin 150040, China

4   Southeast University-University of Wisconsin Intelligent Network Transportation Joint Research Institute, 2312 Engineering Hall, 1415 Engineering Drive, Madison, WI 53706, USA

\*   Correspondence: hymjob@nefu.edu.cn (Y.-M.H.); peiyulong@nefu.edu.cn (Y.-L.P.);
    Tel.: +86-136-3360-2189 (Y.-M.H.); +86-159-4566-5286 (Y.-L.P.)

**Abstract:** To find and solve the problems existing in the development of higher education in China, the input–output, scale of higher education, students' tuition and teachers' income of Chinese and American universities are compared. The results show that the investment in top universities in China is similar to that in the United States, but the average student budget is much less, and the output is not comparable to that of American universities. The scale of higher education is much larger than that of the United States, and the growth rate is far more than demand. College tuition should be increased, with the absolute tuition only 5.93% of income, and relative tuition is 20.21% of that in the United States. College teachers are underpaid, earning only approximately 20% of what their peers earn in the United States. Therefore, for higher education sustainability, the paper puts forward the development direction of higher education in China, which is to control the expansion scale of colleges and universities, and to increase students' tuition and teachers' salary.

**Keywords:** higher education; sustainability American universities; tuition income ratio; educational input; teachers' salaries

---

## 1. Introduction

Since the reform and opening up, higher education has been developed rapidly [1], and some achievements have been made in China [2]. However, compared with Western countries, especially the United States, there is still a significant gap [3]. In fact, universities in China are characterized by low tuition for students, low income for teachers, low efficiency for universities, and high investment from the government [4].

With the expansion of college enrollment, the number of students in China has steadily come to be ranked first in the world, but the rapid expansion of colleges and universities also brings a series of problems [5]. First, higher education is mainly run by the public sector, and the funding is mainly from the government at all levels. Second, due to the low tuition and limited financial expenditure, teachers' salary and research funds are insufficient, resulting in the low income of university teachers. In particular, the income of young teachers in less-developed provincial colleges and universities is very low. Provincial colleges and universities refer to institutions that receive most of their financial support from the provinces in which they are located. Provincial colleges and universities, like those

under the ministry of education, are also public colleges and universities. In China, the best colleges and universities are under the ministry of education, then provincial ones, with the worst being private colleges and universities. The low income level of less-developed provincial colleges and universities means that they are unable to attract outstanding talent [6]. Third, as most expansion of enrollment is occurring in less-developed provincial and private colleges and universities, the overall level of higher education in China has been dragged down to some extent. Fourth, a large number of people of the right age have enrolled in colleges and universities in recent years, leading to a shortage of the labor force, and this problem is becoming increasingly serious [7]. This has had a large impact on the manufacturing industry in the country known as the "factory of the world" [8].

Wen Fan et al. analyzed the public investment and external returns to higher education. The result showed that the returns are positive and statistically significant for rural, male, and poorly educated workers, and China should increase public investment in education and target rural areas and poorly educated workers [9]. Ha Sha focused on some basic issues common to higher education institutions in China and in Italy, such as the degree of autonomy from political power in academic governance and the quality of the knowledge production and transfer to society. Remarkable financial investments from the central and regional authorities in China have paid off, contributing to the technological advancement of the country. Unfortunately, the great financial crisis in Italy caused a strong reduction of public funds to universities and a consequent brain drain of young post graduates in northern Europe and North America [10]. Kivinen O et al. introduced a relational input–output model for the productivity analysis of university research. Their comparative analyses focused on top university research in hard sciences from 4 East Asian countries and 4 northern European countries. The results showed that northern European countries are ranked higher in the "knowledge economy indicators", but East Asians fare better in indicators for learning outcomes and by the productivity of university research in the natural sciences and technology; north European countries are stronger in clinical medicine [11].

Liang Cheng Zhang et al. investigated economies of scale and scope in higher education to offer public and private providers and stakeholders of college and university teaching, research, and other services with knowledge of the cost structures that underpin provision in this economically and socially important sector. Their findings suggest that functional form and allowances for managerial efficiency have a significant impact on the estimated scale economies [12]. Julian L et al. analyzed the higher education tuition in OECD (Organization for Economic Co-operation and Development) countries. They observed large differences between countries. When a student graduates from college in the USA, he/she probably shoulders debt of US $4000–10,000 due to loans taken out to cover tuition fees and living expenses. In many other countries, students graduate with no tuition fees or living expenses [13]. Mingyue Gao analyzed the characteristics of the salary system of American college teachers, and then through comparative analysis, analyzed the weak competitiveness of the salary level of Chinese college teachers in the talent market in China and internationally, the unreasonable salary structure, and the large salary gap within the group. Aiming to address these problems, this paper puts forward measures for the comprehensive reform of the salary and personnel system of college teachers, including strengthening the external competitiveness of the salary, formulating a reasonable salary structure, and improving the salary distribution of colleges and universities [14].

Qiu Yong et al. thought that the construction of a given college depends on the college teachers, and that it is very important to solve the salary problem of college teachers. Therefore, they combined the history of college teachers' salary in China with the experiences of the United States, the UK, Germany, Canada, Japan, India, and China, Hong Kong to analyze this problem. They put forward six suggestions for the reform of the salary system for college teachers in China from the perspective of compensation management [15]. Min Hong sketched university governance in Australia and China and found that the relationship between government and university is looser in Australia than in China. Australian universities enjoy more autonomy and power than Chinese universities; for university internal governance, Australian universities use a more business-oriented management

mechanism [16]. Simon Marginson explored the intersection between stratified education backgrounds and the stratified structures. These different structures include public/private distinctions in schooling and higher education, different fields of study, binary systems and tiered hierarchies of institutions, and the unequal effects of tuition. Larger social inequalities set limits on what education can achieve [17].

The above research can be used for reference in promoting the development of higher education in China. However, the previous studies do not completely describe China's national conditions and the current situation of higher education. Therefore, our project research group proposed the development direction of higher education in China on the basis of comparing Chinese and American universities in terms of input and output, scale of higher education, student tuition, and teacher income.

## 2. Input and Output of Chinese and American Universities

### 2.1. Funding Input of Chinese and American Universities

The Chinese government has implemented a project to build first class universities and first class disciplines. The relevant universities are double first-class universities. According to the budget expenditure of universities directly under the Ministry of Education from 2016 to 2018, our project research group developed statistics and rankings on the budget expenditure data of double first-class universities over the past three years. The top ten are shown in Table 1.

**Table 1.** Budget ranking in the past three years (¥ Billion).

| Rank | Universities | 2016 | 2017 | 2018 | Total |
|------|-------------|------|------|------|-------|
| 1 | Tsinghua University | 13.70 | 19.34 | 21.63 | 54.67 |
| 2 | Peking University | 9.41 | 14.59 | 12.55 | 36.54 |
| 3 | Zhejiang University | 9.00 | 10.81 | 11.40 | 31.20 |
| 4 | Shanghai Jiaotong University | 8.21 | 11.28 | 11.49 | 30.97 |
| 5 | Sun Yat-Sen University | 6.37 | 8.48 | 10.21 | 25.06 |
| 6 | Fudan University | 5.86 | 7.46 | 7.98 | 21.30 |
| 7 | Tianjin University | 4.34 | 9.71 | 6.60 | 20.65 |
| 8 | Tongji University | 4.32 | 5.49 | 10.72 | 20.53 |
| 9 | Wuhan University | 5.43 | 7.25 | 7.05 | 19.73 |
| 10 | Huazhong University of Science and Technology | 5.33 | 6.49 | 7.70 | 19.52 |

Higher education in China is mainly run by the government. For example, in China, the top 200 universities are all public. By contrast, most of the top 200 American universities are private. The funding in China is also mainly from the government at all levels. This is very different from American universities. China is a developing country with limited financial resources, and higher education relying mainly on government support is not sustainable.

Compared with top American universities, there is little difference in the total amount of financial investment. In 2016, the investment to Tsinghua University was 13.7 billion Yuan ($3.57 billion in purchasing power terms), higher than Massachusetts Institute of Technology's $3.34 billion and Yale's $3.36 billion in 2014. Peking University spent $2.45 billion, nearly the same as Caltech and the University of Washington. Zhejiang University and Shanghai Jiaotong University spent US $2.30 billion and US $2.10 billion, respectively. Fudan University, which ranks fifth, spent US $1.50 billion, comparable to Princeton University in the United States. But the average student budget varies widely. Tsinghua University and Peking University, China's no. 1 and no. 2 universities, spend less than a quarter of the U.S. budget per student. The budgets of some Chinese and American universities are shown in Table 2.

**Table 2.** Budgets of some Chinese and American universities.

| Universities | Students | Budgets (Billion) | Budget/Student (US $) |
|---|---|---|---|
| Massachusetts Institute of Technology | 11,376 | 3.34 | 293,601 |
| Yale University | 12,385 | 3.36 | 271,296 |
| Princeton University | 8623 | 1.51 | 175,113 |
| California Institute of Technology | 30,835 | 2.46 | 79,779 |
| Tsinghua University | 48,739 | 3.57 | 73,247 |
| Peking University | 36,305 | 2.45 | 67,484 |
| University of Washington | 54,532 | 2.43 | 44,561 |

As can be seen from the table above, the most average budget of American college students is much higher than that of Chinese students. With Tsinghua University and Peking University, the first and second universities in China, their average budget for each student is less than a quarter of MIT's.

The top income sources of universities in China are similar to those in the United States, most of which are affiliated hospitals, industrial real estate, etc., and a small part comes from government fiscal appropriation. The proportion of fiscal appropriation of universities directly under the Ministry of Education in 2018 is shown in Table 3.

**Table 3.** Proportion of fiscal appropriation in total expenditure (2018) (¥ Billion).

| Rank | | Universities | Total Appropriation | Total Budget | Appropriation Proportion |
|---|---|---|---|---|---|
| Smallest top 5 | 1 | Tongji University | 2.20 | 10.70 | 20.56% |
| | 2 | Tsinghua University | 5.00 | 21.60 | 23.15% |
| | 3 | Sun Yat-Sen University | 2.50 | 10.20 | 24.51% |
| | 4 | Shanghai Jiaotong University | 3.20 | 11.50 | 26.09% |
| | 5 | Zhejiang University | 3.00 | 11.40 | 27.19% |
| Largest top 5 | 5 | Sichuan University | 3.10 | 8.50 | 36.47% |
| | 4 | Hunan University | 1.60 | 4.10 | 39.02% |
| | 3 | Peking University | 5.30 | 12.50 | 42.40% |
| | 2 | Northwest A and F University | 1.60 | 3.70 | 43.24% |
| | 1 | Lanzhou University | 1.70 | 3.70 | 45.95% |

Compared with the universities affiliated with the Ministry of Education, there is an obvious gap in investment between provincial universities and private universities. In 2018, there were 2631 universities in China, among which 735 were private, accounting for 29.94% of the total. The total budget of private universities is only 111.5 billion Yuan, with the average budget per university amounting for less than 200 million Yuan. The main income of provincial and private universities in China comes from tuition and government subsidies.

According to data from the National Center for Education Statistics in USA, as of fall 2019, there were 4724 universities in the United States, among which 2898 were private universities, accounting for 61.35% of the total. The number and proportion of private colleges and universities are more than 2 times the figures in China. Private colleges in the United States are mostly funded by high tuition and private donations. Private universities often have higher budgets than public universities. The top 20 most popular U.S. universities are private. The well-known universities, such as Harvard University, Princeton University, Yale University and Stanford University, are all private universities. In contrast, the top 20 most popular Chinese universities are all public.

*2.2. Output of Chinese and American Universities*

The output evaluation of colleges and universities mainly considers teaching, research, citations, international outlook, and industrial income [18]. The comprehensive calculation is carried out

according to a certain weight. The factors and weights considered in the output evaluation of colleges and universities are shown in Table 4.

**Table 4.** Factors and weights considered.

| Number | Factor | Specific Content | Weight |
|--------|--------|------------------|--------|
| 1 | Teaching | Learning environment | 30.0% |
| 2 | Research | Publication, salary, and reputation | 30.0% |
| 3 | Citations | Study impact | 30.0% |
| 4 | International outlook | Staff, students, and research | 7.5% |
| 5 | Industrial income | Knowledge transformation | 2.5% |

Based on the table above, the *Times Higher Education Edition* published the ranking of world universities in 2019. Seven of the top ten universities are from the United States, and none of them are from China. The top ten universities in the world are listed in Table 5.

**Table 5.** Top 10 universities in the world.

| Rank | Universities | Country | Rank | Universities | Country |
|------|-------------|---------|------|-------------|---------|
| 1 | Oxford University | UK | 6 | Harvard University | USA |
| 2 | University of Cambridge | UK | 7 | Princeton University | USA |
| 3 | Stanford University | USA | 8 | Yale University | USA |
| 4 | Massachusetts Institute of Technology | USA | 9 | Imperial College London | UK |
| 5 | California Institute of Technology | USA | 10 | University of Chicago | USA |

As can be seen from the table above, the top 10 universities are British and American. They have one thing in common, that is, they are in English-speaking countries. Universities are ranked by *Times Higher Education Edition* on the basis of teaching effectiveness, research results, and international impact, most of which are presented in English, giving British and American universities strong advantages in ranking. On the contrary, universities in Germany, Japan, China, and other non-English-speaking countries are at a disadvantage in ranking.

In China, only 6 universities made the top 100. They are Tsinghua University, Peking University, University of Science and Technology of China, University of Hong Kong, Hong Kong University of Science and Technology, and Chinese University of Hong Kong. The Chinese universities listed in the *Times Higher Education Edition* top 100 world university rankings in 2019 are presented in Table 6.

**Table 6.** World University Ranking in 2019 (Chinese universities).

| Universities | Rank | Universities | Rank |
|-------------|------|-------------|------|
| Tsinghua University | 22 | Hong Kong University | 36 |
| Peking University | 31 | Hong Kong University of Science and Technology | 46 |
| University of Science and Technology | 93 | Chinese University of Hong Kong | 55 |

It is notable that Tsinghua University rose from 30th in 2018 to 22nd in 2019 and replaced the National University of Singapore as the top university in Asia. Zhejiang University (105), Fudan University (105), and the Chinese University of Hong Kong (110) have the potential to enter the top 100.

As shown in Tables 6 and 7, the rankings of Tsinghua University and Peking University are steadily rising, while the rankings of the three universities in Hong Kong have declined to different degrees, and Taiwan University has disappeared from the list. As a microcosm of the development of universities in mainland China, the progress of Tsinghua University and Peking University represents the progress of universities in China.

**Table 7.** World University Ranking in 2009 (Chinese Universities).

| Universities | Rank | Universities | Rank |
|---|---|---|---|
| Tsinghua University | 49 | Hong Kong University | 24 |
| Peking University | 52 | Hong Kong University of Science and Technology | 35 |
| Taiwan University | 95 | Chinese University of Hong Kong | 46 |

*2.3. Comparison of Input and Output between Chinese and American Universities*

The Chinese government attaches great importance to and invests much in higher education. However, because China's economic development level is much lower than that of the United States, the investment in colleges and universities cannot be compared with that of the United States. The main reason for the huge gap in output between Chinese and American universities is the lack of investment in Chinese universities.

The second reason is historical. The average history of Chinese colleges and universities is shorter than 40 years, and all colleges and universities founded before the founding of the People's Republic of China have experienced the baptism of war and ten years of civil unrest during the Cultural Revolution. In contrast, the average history of American colleges and universities is longer than 70 years, and most have not experienced war or civil unrest.

However, judging from the development trend of the last decade, Chinese universities are making great strides forward and the gap with the United States is continually shrinking.

## 3. Higher Education Scale in China and the United States

*3.1. Higher Education Scale in China*

The number of colleges and universities in China has increased rapidly in the past 20 years. In 1998, there were only 1022 colleges and universities in China, among which 590 were undergraduate colleges and 432 were vocational and technical colleges. In 1999, universities began to expand enrollment on a large scale, and the number of universities increased rapidly. By the end of 2018, there were 2631 national ordinary universities. Although the total number of colleges and universities in China has already doubled, most of the newly added undergraduate course colleges and universities are the result of upgrading of the original higher vocational colleges. The newly added higher vocational colleges are mostly upgraded from newly established private colleges or secondary schools. According to statistics, there are only approximately 600 universities with more than 20 years of experience in running a university. The expansion of enrollment in the past 20 years has brought down the overall level of higher education in China.

Since the restoration of the college entrance examination system in 1990, except for some years, the number of students has been on the rise. The number of students enrolled in 2018 reached a record 7.91 million, with an admission rate of 81.13%. The numbers of new students enrolled in Chinese colleges and universities in the recent 30 years are listed in Table 8.

As shown in Table 8, new student admissions in 2018 are almost 13 times that of 1989. In comparison, from 1989 to 2018, China's population increased from 1.12 billion to 1.39 billion, only increasing by 24.11%. In the past 30 years, the massification of higher education in China has developed rapidly, as has also happened in many other countries. The enrolment rates of college students of college age in the past 10 years in China are listed in Table 9.

Due to the one-child policy in China, there are fewer and fewer young people of college age, but the number of college admission of new students is increasing, so the admission rate has doubled over the 10 years listed. According to the National Education Development Report released by the Ministry of Education in October 2018, there are 37.79 million students in all types of higher education in China. The gross enrollment rate of higher education reached 48.12%, and a number of super universities were formed, with 60 or 70 thousand students.

**Table 8.** Admissions over the years in China (Million).

| Years | Admissions | Years | Admissions | Years | Admissions |
|---|---|---|---|---|---|
| 1989 | 0.59 | 1999 | 1.60 | 2009 | 6.29 |
| 1990 | 0.60 | 2000 | 1.80 | 2010 | 6.57 |
| 1991 | 0.62 | 2001 | 2.68 | 2011 | 6.75 |
| 1992 | 0.75 | 2002 | 3.21 | 2012 | 6.85 |
| 1993 | 0.92 | 2003 | 3.82 | 2013 | 6.84 |
| 1994 | 0.90 | 2004 | 4.47 | 2014 | 6.98 |
| 1995 | 0.93 | 2005 | 5.05 | 2015 | 7.00 |
| 1996 | 0.97 | 2006 | 5.40 | 2016 | 7.72 |
| 1997 | 1.00 | 2007 | 5.67 | 2017 | 7.00 |
| 1998 | 1.08 | 2008 | 6.08 | 2018 | 7.91 |

**Table 9.** Enrolment rates in the recent 10 years in China.

| Years | New Students Admissions (Million) | Students of College Age (Million) | Rates (%) |
|---|---|---|---|
| 2009 | 6.29 | 25.93 | 24.26 |
| 2010 | 6.57 | 24.73 | 26.57 |
| 2011 | 6.75 | 25.05 | 26.95 |
| 2012 | 6.85 | 22.66 | 30.23 |
| 2013 | 6.84 | 19.80 | 34.54 |
| 2014 | 6.98 | 18.60 | 37.52 |
| 2015 | 7.00 | 17.49 | 40.02 |
| 2016 | 7.72 | 18.06 | 42.74 |
| 2017 | 7.00 | 15.32 | 45.70 |
| 2018 | 7.91 | 16.44 | 48.12 |

Zhengzhou University, Jilin University, and Huazhong University of Science and Technology are the three universities with the most students in China in 2018. The top 10 universities by the number of students in China are shown in Table 10.

**Table 10.** The top 10 universities with the most students in China.

| Rank | Universities | Students (Thousand) | Rank | Universities | Students (Thousand) |
|---|---|---|---|---|---|
| 1 | Zhengzhou University | 72.60 | 6 | Shandong University | 59.50 |
| 2 | Jilin University | 69.60 | 7 | Wuhan University | 56.80 |
| 3 | Huazhong University of Science and Technology | 61.70 | 8 | Central South University | 55.70 |
| 4 | Sichuan University | 60.60 | 9 | Wuhan University of Technology | 54.00 |
| 5 | Henan University | 60.40 | 10 | Harbin Institute of Technology | 53.30 |

Note: the number of students on campus includes those with bachelors, masters, and doctoral degrees.

The expansion of college enrollment has extended the opportunity to receive higher education to many young people of the right age. The shortage of teachers, the poor condition of buildings and other hardware facilities, and the unsound management system are the major problems existing in most provincial colleges and vocational colleges.

These are far from the biggest problems caused by the expansion of college enrollment. On the one hand, many college students cannot find jobs and a large number of graduates are unemployed. On the other hand, manufacturing plants and construction sites cannot find workers even at high wages. There is a strange phenomenon that college graduates earn far less than factory workers and construction workers. The unemployment rate and job income of Chinese college graduates are shown in Tables 11 and 12.

**Table 11.** Unemployment rate of Chinese college graduates (2018).

| Discipline | Unemployment Rate (%) | Discipline | Unemployment Rate (%) | Discipline | Unemployment Rate (%) |
|---|---|---|---|---|---|
| Engineering | 6.9 | Economics | 9.4 | Art | 12.7 |
| Management | 7.3 | Agriculture | 9.6 | History | 13.9 |
| Education | 9.2 | Science | 10.1 | Law | 14.9 |
| Medicine | 9.3 | Literature | 10.5 | **Average** | **10.3** |

**Table 12.** Comparison of the average (2018).

| Building Workers | | Factory Workers | | College Graduates | |
|---|---|---|---|---|---|
| Type of Work | Income | Type of Work | Income | Discipline | Income |
| Carpenters | 94,860 | Clothes-makers | 74,256 | Engineering | 84,532 |
| Electricians | 95,388 | Shoemaking workers | 87,692 | Education | 55,729 |
| Plumbers | 80,465 | Electronics factory workers | 67,345 | Management | 67,491 |
| Brick layers | 110,432 | Toy workers | 73,426 | Medicine | 57,920 |
| Average | 95,286 | — | 75,680 | — | 66,418 |

As can be seen from the above table, the average unemployment rate of Chinese college graduates is 10.3%. According to the data released by the Ministry of Human Resources and Social Security, the average unemployment rate of the urban population in China is 4.9%. That means the unemployment rate for college graduates is more than twice the national average.

As can be seen from the table above, college graduates earn much less than factory workers as well as construction workers. In general, construction and factory workers are much less educated than college graduates. As examined from another angle, there is an oversupply of university graduates in China.

*3.2. Higher Education Scale in the USA*

According to the National Center for Education Statistics, as of fall 2019, there were 4724 colleges and universities in the United States, including 2898 private ones.

There were 2.9 million new college students admitted in the fall of 2019, attending 4-year (1.29 million in public colleges and 0.59 million in private colleges) and 2-year programs (approximately 0.95 million in public colleges and 0.05 million in private colleges). The number of new college students is expected to remain steady, at approximately 3 million by 2030. The admissions scale in 2018 is list in Table 13.

**Table 13.** Admissions scale in 2018 (Million).

| Admissions | 4-Year | 2-Year | Total |
|---|---|---|---|
| Public | 1.29 | 0.59 | 1.88 |
| Private | 0.95 | 0.05 | 1.00 |
| Total | 2.24 | 0.64 | 2.88 |

There are a large number of foreign students among American college freshmen. At the peak in 2016, there were 245,800 foreign students, including 119,300 undergraduates and 126,500 postgraduates. In the past two years, the number of foreign students has begun to decline. In 2018, a total of 226,500 foreign students were admitted, including 108,500 undergraduates and 111,800 postgraduates.

According to the national center for education, college enrollment in the United States has generally been on the rise, reaching a record high of 21.02 million in 2010 and then gradually declining to 19.66 million in 2017. Enrollment in 2018 rose to 19.83 million. Enrollments over the years in the USA are shown in Table 14.

**Table 14.** Enrollments over the years in the USA (million).

| Years | Enrollments | Years | Enrollments | Years | Enrollments |
|-------|-------------|-------|-------------|-------|-------------|
| 1989 | 13.54 | 1999 | 14.85 | 2009 | 20.31 |
| 1990 | 13.81 | 2000 | 15.31 | 2010 | 21.02 |
| 1991 | 14.36 | 2001 | 15.92 | 2011 | 21.01 |
| 1992 | 14.48 | 2002 | 16.61 | 2012 | 20.64 |
| 1993 | 14.31 | 2003 | 16.91 | 2013 | 20.38 |
| 1994 | 14.28 | 2004 | 17.27 | 2014 | 20.2 |
| 1995 | 14.26 | 2005 | 17.49 | 2015 | 19.99 |
| 1996 | 14.37 | 2006 | 17.76 | 2016 | 19.85 |
| 1997 | 14.51 | 2007 | 18.25 | 2017 | 19.66 |
| 1998 | 14.51 | 2008 | 19.10 | 2018 | 19.83 |

According to the 2026 Education Statistics Forecast released by the National Center for Education Statistics (NCES) in 2018, the total number of college students in the United States in 2017 was 19.66 million, and the number was predicted to reach 22.6 million in 2026. The University of Central Florida, Texas A&M University, and Florida International University have the largest number of students. The top 10 American universities with the most students are shown in Table 15.

**Table 15.** The top ten universities with the largest number of students.

| Rank | Universities | Students (Thousand) | Rank | Universities | Students (Thousand) |
|------|-------------|--------------------|------|-------------|--------------------|
| 1 | University of Central Florida | 57.00 | 6 | Arizona State University | 42.40 |
| 2 | Texas A&M University | 53.10 | 7 | University of Pennsylvania | 40.80 |
| 3 | Florida International University | 47.60 | 8 | Blair College | 40.80 |
| 4 | Ohio State University | 46.00 | 9 | University of Texas at Austin | 40.50 |
| 5 | Liberty University | 45.80 | 10 | Michigan State University | 39.00 |

The United States is still the only superpower in the world, and higher education has developed at a high level. Talents from all over the world are attracted to the United States. Overall statistics on international students show that there were 1,078,822 international students studying in the United States, including 35,755 Chinese students, nearly one-third of the total in the 2017–2018 academic year.

*3.3. Comparative Analysis of the Admission Scale of Chinese and American Universities*

In 2018, Chinese universities admitted 7.91 million new students, accounting for 5.69 of the total population of 1.39 billion, and enrolled 37.79 million students, accounting for 27.19 of the total population. The United States admitted 4.45 million new students, accounting for 13.48 of the total population of 330 million, and enrolled 20.14 million students, accounting for 61.03 of the total student-age population. Compared with the United States, higher education admission and enrolment in China are not large, accounting for less than half of the figures for the United States. The problem is that the number of college students in China is growing too fast. The number of college students from 1977 to 2018 is shown in Figure 1.

According to Figure 1 and Table 8, college admission in China expanded the fastest from 1999 to 2009, increasing from 1.6 million to 6.29 million, a level 3.93 times that in 1999, and presenting an average annual growth rate of 14.67%.

American college admission is also growing, but only at an average annual rate of 1.29%. Therefore, although China has a comparatively small number of students, the growth rate is very fast, more than 10 times that of the United States.

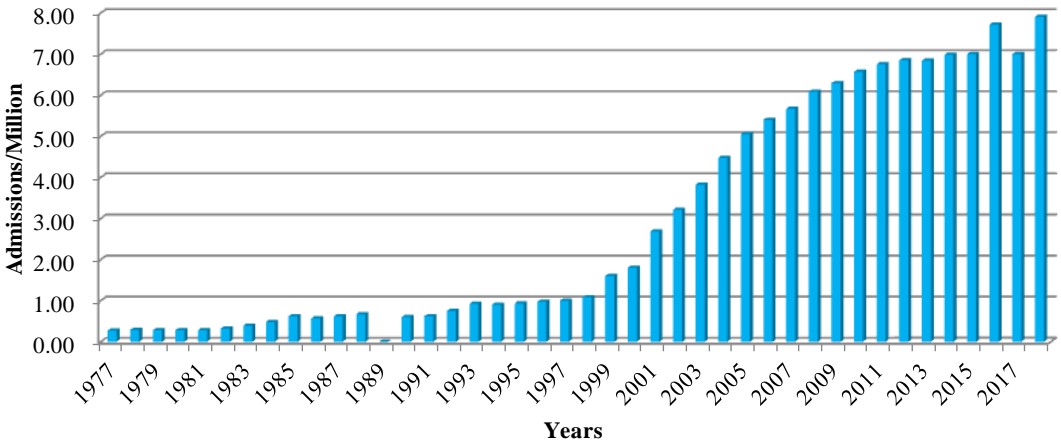

**Figure 1.** The admission of new students.

There are some similarities between the expansion of Chinese and American universities, but China's problems are far more serious. The expansion of higher education in the United States took place in the 1960s and 1970s. After the expansion, community colleges sprung to life in the United States, with low admission standards, low teacher salaries, and little scientific achievement, making them similar to vocational and technical colleges in China. After the expansion in China, there are nearly 2700 universities, but more than 2000 are vocational and technical colleges and only 600 undergraduate schools. However, the difference is that the tuition fees of vocational and technical colleges in China are higher than those of undergraduate universities, which are generally about twice as high as those of undergraduate universities. In addition, the staff in vocational and technical colleges in China are all full-time, and the government needs to pay full salaries, so the expansion of universities has added a great burden to the government. The problems caused by the rapid expansion of universities in China are troubling. We are soberly aware that China still needs to rely on manufacturing for a long time, and the knowledge economy is still far behind that of the United States. The labor shortage caused by the expansion of Chinese universities has taught the Chinese government a lesson, with manufacturing costs rising and many manufacturing plants already moving to Southeast Asia. The expansion of Chinese universities has contributed to the reduction in GDP (Gross Domestic Product) growth from a peak of 15% to about 6% today.

## 4. Comparison of Tuition between Chinese and American Universities

### 4.1. Tuition of Chinese Universities

When the college entrance examination was reinstated in 1978, Chinese universities did not charge tuition, and even food and housing were paid by the government. In 1985, a "dual-track system" was implemented for college enrollment. The planned enrollment without charge and the regulated enrollment with charge coexisted. From 1988 to 1994, all universities in China charged tuition amounting to only a nominal 200 Yuan per year. Following the trial implementation of "parallel track" enrollment in 1994, college tuition began to gradually increase. In 1999, college enrollment expanded, and tuition rose rapidly in the following years. At present, the tuition for an ordinary major in colleges and universities is approximately 5000~6000 Yuan per year, and that for an art major is generally more than 10,000 Yuan. The average tuition of the top ten universities in 2018 surveyed by the project team is shown in Table 16.

**Table 16.** Average undergraduate tuition of the top 10 universities in China.

| Rank | Universities | Tuition (¥/Year) | Rank | Universities | Tuition (¥/Year) |
|---|---|---|---|---|---|
| 1 | Tsinghua University | 5154 | 6 | Zhejiang University | 5964 |
| 2 | Peking University | 5095 | 7 | Nanjing University | 5468 |
| 3 | University of Science and Technology of China | 4940 | 8 | Sun Yat-Sen University | 5947 |
| 4 | Fudan University | 5846 | 9 | Huazhong University of Science and Technology | 5836 |
| 5 | Shanghai Jiao Tong University | 5509 | 10 | Harbin Institute of Technology | 4894 |

According to Table 16, the average tuition of the top ten universities in China can be calculated to be 5465 Yuan. Tuition at Chinese universities is set by the government and are only related to the major, private, or public, regardless of the region. Tuition at Chinese top universities is usually about half of that of vocational and technical colleges, so the average tuition at Chinese universities is about 11,000 Yuan. The statistics from the National Bureau of Statistics show that the average annual income of in-service employees in China was 90,960 Yuan in 2018. According to this calculation, the average tuition accounts for only 12.09% that of the average annual income.

*4.2. Tuition of American Universities*

Tuition at American universities is generally more expensive, the tuition of top universities is typically around $50,000, and it increases by a certain percentage every year. Schools adjust tuition according to the income and expenditure of the school and the charging standards of similar universities. According to the U.S. College Board, tuition for private universities has been rising in a straight line every year. From 2006 to 2016, the average annual tuition fees of private universities increased by 2.4%. The average undergraduate tuition of the top 10 U.S. universities surveyed by the project group in the fall of 2018 is listed in Table 17.

**Table 17.** Average undergraduate tuition of the top 10 universities in the USA.

| Rank | Universities | Tuitions ($/Year) | Rank | Universities | Tuitions ($/Year) |
|---|---|---|---|---|---|
| 1 | Princeton University | 45,320 | 6 | Yale University | 53,430 |
| 2 | Harvard University | 47,074 | 7 | Stanford University | 47,940 |
| 3 | Columbia University | 59,430 | 8 | Duke University | 55,960 |
| 4 | Massachusetts Institute of Technology | 51,832 | 9 | University of Pennsylvania | 55,584 |
| 5 | University of Chicago | 57,006 | 10 | Johns Hopkins University | 50,410 |

According to Table 17, the average cost of tuition fees at the top 10 universities in the United States is $52,399. Considering that the average tuition at state and community colleges in the United States is relatively low, at about $19,000 a year, the average annual tuition at colleges across the country is about $26,000. Statistics from the Bureau of Labor Statistics show that the average salary of employees in the United States is $43,460. Average tuition fees account for up to 59.83% of average annual income.

*4.3. Comparative Analyses of Tuition Fees in Chinese and American Universities*

The research team used relative and absolute tuition fees to compare tuition fees in Chinese and American universities. Relative tuition refers to the comparison between domestic tuition fees and domestic income. We introduced the definition of "tuition income ratio", that is, the percentage of tuition fees of the average annual income of on-the-job employees. The above analysis shows that the average tuition income ratio of all the universities in China is 12.09%, while the average tuition income ratio of the universities in the United States is 59.83%. It means that the tuition income ratio of Chinese universities is only 20.21% that of the United States.

Absolute tuition refers to the conversion of tuition from one country to the currency of another country to facilitate the comparison of tuition from different countries. The research team also converted the tuition of Chinese universities into US dollars. The average tuition of the top 10 universities in China was US $859 ($1 = ¥7.14), which was only 1.64% of the top 10 universities in the United States, and it is US $52,399 in the same year. The average absolute tuition in Chinese universities is $1540.62, compared to $26,000 in the United States. The average absolute tuition of Chinese universities is only 5.93% of that of the United States.

The analysis above shows that college tuition in China is at a "very low" level compared with the tuition in the United States. Therefore, it is suggested to gradually increase tuition on the basis of sufficient research and use the income from the increased tuition to subsidize poor students and increase scholarships and teachers' income.

## 5. Chinese and American College Teachers' Income Comparison

### 5.1. Chinese College Teachers' Income

In 2013, the Salary Management Research Branch of the Chinese Society of Higher Education surveyed the income level of Chinese university teachers. More than 130,000 university teachers and 84 universities were included [19]. The results showed that the average annual income of all respondents was 113,500 Yuan. The annual income of Chinese college teachers and young teachers under 40 years old is shown in Figures 2 and 3.

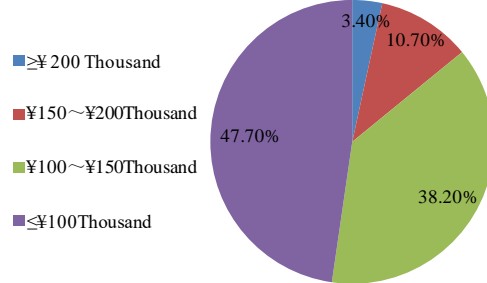

**Figure 2.** Annual income of teachers.

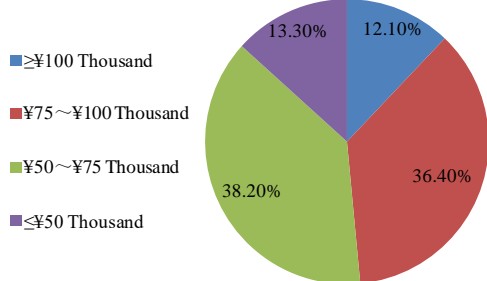

**Figure 3.** Annual incomes of young teachers.

Figure 2 shows that nearly half of college teachers have an annual income less than 100,000 Yuan, and those whose incomes are between 100,000 and 150,000 Yuan account for nearly 40%, while only 3.4% of college teachers have an annual income of more than 200,000 Yuan. As shown in Figure 3, the income of young teachers is even lower. Up to 81.9% of young teachers have an annual income of less than 100,000 Yuan, which is low for most young teachers with masters or doctoral degrees.

In the same year, the China Federation of Trade Unions reported average salaries across industries, showing that bankers had the highest average earnings. The salary of college teachers ranked 10th among the 36 major industries, and the income data were similar to the survey results. The income of the top ten industries is shown in Table 18.

**Table 18.** The income of the top ten industries in China.

| Rank | Occupation | Average Income (Yuan/Year) | Rank | Occupation | Average Income (Yuan/Year) |
|------|-----------|----------------------------|------|-----------|----------------------------|
| 1 | Banking | 340,325 | 6 | Real estate | 156,345 |
| 2 | IT | 281,027 | 7 | Tourism | 146,459 |
| 3 | Lawyer | 223,422 | 8 | Transportation | 132,657 |
| 4 | Architecture | 187,432 | 9 | Government employees | 128,735 |
| 5 | Doctor | 168,036 | 10 | University Teachers | 115,384 |

As can be seen from the table above, university teachers earn far less than bankers and IT workers, lawyers, and doctors, and even less than government employees. And the average education level of practitioners in these high-income industries is not as good as that of college teachers.

In recent years, with the development of the economy, the salary level of college teachers has been increasing year by year, and the salary composition has been changing. At present, colleges and universities in China are implementing the basic salary and performance salary system. Teachers' income mainly includes a basic salary based on professional title and a performance salary based on teaching and research output. The research group took Northeast Forestry University as an example to annualize the salary income of teachers in colleges and universities in China. Northeast Forestry University is a "211 project" key university directly under the administration of the Ministry of Education. "211 project" refers to build about 100 key universities oriented towards the 21st century in China. The income of teachers in Northeast Forestry University mainly includes the basic salary and performance salary. Among them, the basic salary is related to professional title and length of service, while the performance salary is related to professional title and output. The mean salary of teachers in Northeast Forestry University is shown in Table 19.

**Table 19.** Mean salary of teachers in Northeast Forestry University (Yuan).

| Position | Senior | | | | Deputy Senior | Intermediate | Junior | Others |
|----------|--------|--------|--------|--------|---------------|--------------|--------|--------|
| Position Grade | I | II | III | IV | V~VII | VIII~X | XI,XII | XIII |
| Basic salary | 24,466 | 8327 | 7658 | 5376 | 4876 | 3958 | 3254 | 2543 |
| Performance salary | 20,000 | 7600 | 6300 | 5500 | 4600 | 3600 | 2700 | 2200 |
| Total (Month) | 44,466 | 15,927 | 13,958 | 10,876 | 9476 | 7558 | 5954 | 4743 |
| Total (Year) | 533,592 | 191,124 | 167,496 | 130,512 | 113,712 | 90,696 | 71,448 | 56,916 |

The average basic salary in the table above is provided by the finance department of Northeast Forestry University. According to the distribution method of post subsidies of Northeast Forestry University in 2018, all of the performance salary is directly allocated to the college by the university. The table above presents the average value of allocation. It also shows that for young teachers to obtain an average income of more than 100,000 Yuan, they must reach the deputy title or higher, and title promotion is not easy.

With the replacement of old college teachers, young teachers have gradually become the mainstay of teaching and scientific research. The "Green Pepper" was once the focus of social attention. A "Green Pepper" refers to a young college teacher, whose education was high, income was low, and whose work was full of high pressure. Unfortunately, the low income level of young teachers in colleges and universities remains unchanged.

The income gap of Chinese college teachers is very obvious. For example, teachers in the top 10 universities earn far more than the average. The salaries of the top 10 Chinese college teachers are list in Table 20.

**Table 20.** The top ten average salaries of Chinese college teachers (Thousand Yuan).

| Rank | Universities | Full Professor | Associate Professor | Assistant Professor | Average |
|------|-------------|----------------|---------------------|---------------------|---------|
| 1 | Tsinghua University | 278.75 | 185.59 | 153.01 | 205.76 |
| 2 | Beijing University | 276.55 | 156.98 | 139.23 | 190.92 |
| 3 | China University of Science and Technology | 272.36 | 162.09 | 158.76 | 197.74 |
| 4 | Fudan University | 258.65 | 148.67 | 145.91 | 184.39 |
| 5 | Shanghai Jiaotong University | 256.88 | 174.52 | 134.97 | 188.79 |
| 6 | Zhejiang University | 255.25 | 155.49 | 126.66 | 179.13 |
| 7 | Nanjing University | 254.89 | 158.33 | 156.48 | 189.85 |
| 8 | Sun Yat-sen University | 248.07 | 166.92 | 146.83 | 187.30 |
| 9 | Huazhong University of Science and Technology | 246.02 | 154.57 | 121.27 | 173.95 |
| 10 | Harbin Institute of Technology | 241.12 | 146.12 | 136.18 | 174.45 |
| | Average | 258.87 | 160.96 | 141.93 | 187.23 |

Salaries for teachers at China's top universities are higher than not only the national average but also some of China's top 10 industries. Unfortunately, the salaries of top universities cannot represent the overall level of universities in China. The salary level of teachers in Chinese universities is still very low, as discussed above.

The income of Chinese college teachers is also influenced by the prestige, public/private status, the cost of living in the region, the discipline, and so on. Generally speaking, at public universities with good social prestige, the teachers have a high salary. Where the cost of living is high, college teachers are well paid. The highest-paid discipline for Chinese college teachers is banking.

*5.2. American College Teachers' Income*

According to statistics from the American Association of University Professors, college teachers in the United States earned approximately $90,000 in 2018. Among them, the average annual income of full professors, associate professors, and assistant professors was approximately $120,000, $80,000, and $70,000, respectively. The incomes of American college teachers in 2018 are shown in Table 21.

**Table 21.** Income of American college teachers in 2018 (US $).

| Type | Full Professor | Associate Professor | Assistant Professor | Average |
|------|----------------|---------------------|---------------------|---------|
| Public | 113,738 | 81,969 | 70,246 | 88,651 |
| Private | 120,977 | 81,040 | 67,147 | 89,721 |
| Average | 117,358 | 81,505 | 68,697 | 89,186 |

Top universities in America have set up a competitive pay system to attract the best talent from all over the world. The salary of teachers mainly depends on the relationship between the supply and demand of talent, and the income of teachers in the United States is much higher than that of teachers in universities in China. The average salaries of college teachers in the top 10 American universities are shown in Table 22.

The salaries of teachers in American colleges and universities depend not only on their titles and the schools they work for, but also depend on their discipline and the region in which they work. Because the income of different disciplines is related to the industry, and the area where the university is located has a direct impact on the living cost of university teachers [20]. Incomes of university teachers of different disciplines in the United States are list in Table 23.

**Table 22.** The top ten average salaries of American college teachers (US $ Thousand).

| Rank | Universities | Full Professor | Associate Professor | Assistant Professor | Average |
|---|---|---|---|---|---|
| 1 | Stanford University | 280.40 | 186.70 | 153.90 | 207.00 |
| 2 | University of Chicago | 278.20 | 157.90 | 140.10 | 192.10 |
| 3 | Harvard University | 274.00 | 163.10 | 159.70 | 198.90 |
| 4 | New York University | 260.20 | 149.60 | 146.80 | 185.50 |
| 5 | Columbia University | 258.40 | 175.60 | 135.80 | 189.90 |
| 6 | Yale University | 256.80 | 156.40 | 127.40 | 180.20 |
| 7 | University of Pennsylvania | 256.40 | 159.30 | 157.40 | 191.00 |
| 8 | Massachusetts Institute of Technology | 249.60 | 167.90 | 147.70 | 188.40 |
| 9 | Princeton University | 247.50 | 155.50 | 122.00 | 175.00 |
| 10 | Northwestern University | 242.60 | 147.00 | 137.00 | 175.50 |
| | Average | 260.41 | 161.90 | 142.78 | 188.35 |

Note: The data come from startclass.com.

**Table 23.** Income of university teachers of different disciplines in USA (US $ Thousand).

| Rank | Disciplines | Salaries | Rank | Disciplines | Salaries |
|---|---|---|---|---|---|
| 1 | Medical science | 194.90 | 6 | Engineering | 158.10 |
| 2 | Social Science | 176.60 | 7 | Media and Communication | 150.10 |
| 3 | Computer Science | 173.50 | 8 | Humanities | 143.60 |
| 4 | Business | 168.20 | 9 | Fine Arts | 109.60 |
| 5 | Physics and Life Science | 162.60 | 10 | Education | 79.00 |

As can be seen from the table above, there is a great difference in the income of teachers of different disciplines in universities, the highest of which is $194,900 for medicine, and the lowest of which is only $79,000 for education.

The income of American college teachers is closely related to their contribution. First, American universities attach great importance to the performance of teachers and take their contribution and work performance as the basis for the evaluation of position, rank, and salary. Second, due to the fierce competition among colleges and universities for talent, the personnel departments of colleges and universities pay close attention to the salary dynamics, regularly carry out salary surveys within and outside the industry, and serve as the basis for formulating the salary standards of the university.

That is why salaries at top universities and colleges in the United States are so attractive compared with other industries and other institutions of higher education. Teachers in the top universities in the United States have enviable salaries and benefits, which provide a strong guarantee for all kinds of talents to devote themselves to teaching and research. In addition, the fair, reasonable, and scientific salary fluctuation of American universities provides an effective guarantee for the orderly flow of talents.

*5.3. Comparative Analyses on the Salary in Chinese and American Universities*

It is difficult to obtain the average income of national college teachers, and the income of Northeast Forestry University teachers is in the middle and upper level among national colleges and universities. Therefore, the income of teachers in Northeast Forestry University is used to replace the average income of national college teachers for comparison. The professional title system of teachers in American universities is different from that in China; there is no "lecturer" position. In China, "lecturers" and "teaching assistants" are equivalent to "assistant professors" in the United States. Therefore, the average salaries of lecturers and teaching assistants are compared with the salary of assistant professors in the United States. In addition, according to the survey, professors from levels 1 to 4 account for 1%,

4%, 40%, and 55%, respectively. According to this calculation, the average annual income of professors is 151,761 Yuan. Finally, the salary of university teachers is converted into US dollars, as shown in Table 24.

**Table 24.** Comparison of the annual income of Chinese and American college teachers (US $).

| Title | Full Professor | Associate Professor | Assistant Professor |
|---|---|---|---|
| America | 117,358 | 81,505 | 68,697 |
| China | 22,516 | 16,871 | 12,028 |
| Account for (%) | 19.19 | 20.70 | 17.51 |

As shown in Table 24, the salary of college teachers with the same title in China is only approximately 20% of that in the United States. Therefore, the income of college teachers in China is at a "very low" level compared with that in the United States. Only by substantially increasing the income of teachers will it be possible to attract more talents to teach in Chinese universities.

After comparing the average annual income of teachers in Chinese and American universities, let us compare the average annual income of teachers in top universities. We converted the teachers' average income of top universities into dollars before the comparison. Comparison of the top ten universities' average salaries is shown in Table 25.

**Table 25.** Comparison of the top ten universities' average salaries (US $ Thousand).

| Title | Full Professor | Associate Professor | Assistant Professor | Average |
|---|---|---|---|---|
| America | 260.41 | 161.90 | 142.78 | 188.35 |
| China | 43.26 | 39.54 | 26.88 | 36.56 |
| Account for (%) | 16.61 | 24.42 | 18.83 | 19.41 |

As can be seen from the table above, the salary of teachers in China's top universities is also very low compared with that in America, only about 20%. Considering that the average income in China (90,960 Yuan = US $12,739) is 29.31% of the average income in the United States ($43,460), both the average salary of Chinese university teachers and that of top university teachers need to be increased.

*5.4. Salaries of Teachers in Colleges and Universities Worldwide*

In April 2012, the International Higher Education Research Center of the University of Chicago surveyed the incomes of university teachers from 28 countries (Altbach and Pacheco, 2012). According to the survey, new college teachers in China earned $259 per month in terms of purchasing power parity, the third lowest in the world, and the national average income of college teachers was only $720 [21]. The survey results also showed that the lowest and average incomes of Canadian university teachers were ranked first, amounting for 22 times and 10 times the incomes of Chinese university teachers [22]. The results published by the International Higher Education Research Center are shown in Table 26.

As shown in Table 26, the salary level of college teachers in China is far lower than that of developed countries, even developing countries. Therefore, it is imperative to improve the income level of our university teachers.

**Table 26.** Average monthly income of teachers in selected countries (US $).

| Rank | | Countries | Lowest | Highest | Average |
|---|---|---|---|---|---|
| Top 5 | 1 | Canada | 5733 | 9485 | 7196 |
| | 2 | Italy | 3525 | 9118 | 6955 |
| | 3 | South Africa | 3927 | 9330 | 6531 |
| | 4 | India | 3954 | 7433 | 6070 |
| | 5 | United States | 4950 | 7358 | 6054 |
| Bottom 5 | 5 | Kazakhstan | 1037 | 2304 | 1553 |
| | 4 | Ethiopia | 864 | 1580 | 1207 |
| | 3 | China | 259 | 1107 | 720 |
| | 2 | Russia | 433 | 910 | 617 |
| | 1 | Armenia | 405 | 665 | 538 |

Note: The dollar here refers to the "purchasing power evaluation currency", that is, the value of the per capita income of each country converted into the purchasing power of the dollar as the benchmark value. It essentially indicates the purchasing power of this wage in the United States.

## 6. Conclusions

Through a comparative study of Chinese and American higher education, our team reached the following conclusions:

Universities in China have developed rapidly, but there is a large gap compared with universities in the United States. The investment in top universities is not much different from that in the United States, but the average student budget is much less, and the output cannot be compared with American universities. Therefore, for higher education sustainability, there is an urgent need to improve the efficiency of the use of funds and higher education output.

The absolute scale of higher education in China has consistently ranked first in the world. The absolute number of admission and enrollment of students far exceeds the figures of the United States. However, the growth rate is far more than demand, and it brings some problems. So it is necessary to properly control the expansion of the scale of higher education.

Compared with that in the United States, the tuition of Chinese universities is very low; the relative tuition is only 20.21% and the absolute tuition is only 5.93%. Therefore, it is suggested to gradually increase tuition on the basis of sufficient research.

The income of Chinese college teachers is very low, far lower than that of college teachers in developed countries and developing countries, amounting to only approximately 20% of the income of college teachers in the United States. National average income in China is 29.31% of the average income in the United States. The average income of Chinese college teachers' percentage of American college teachers is much less than the average income of Chinese national percentage of the United States. Therefore, it is imperative to improve the income level of college teachers in China.

Our findings are fully consistent with the actions of the Chinese government. In recent years, the Chinese government has begun to control the expansion of higher education, improve research funds efficiency and the output of higher education, and raise the incomes of university teachers. However, as China is still a developing country, some families cannot afford to pay significant tuition fees. Therefore, our next research goal is to help the government formulate policies that cannot only raise tuition fees to maintain the sustainable development of higher education, but also guarantee the enrollment of low-income students.

**Author Contributions:** Conceptualization, Y.-M.H., Y.-L.P. and B.R.; data curation, J.K. and Y.-T.S.; Formal analysis, Y.-M.H., K.J. and Y.-T.S.; Funding acquisition, Y.-M.H. and Y.-L.P.; Investigation, Y.-M.H., K.J. and Y.-T.S.; Methodology, B.R. and Y.-L.P.; Project administration, Y.-M.H. and K.J.; Supervision, B.R. and Y.-L.P.; Validation, Y.-M.H., B.R. and Y.-L.P.; Writing—original draft, Y.-M.H. and K.J.; Writing—review & editing, B.R. and Y.-L.P. All authors have read and agreed to the published version of the manuscript.

**Funding:** We are thankful to the Education Department of Heilongjiang Province, China, for providing financial support with Key Project of Education Science Planning of Heilongjiang Province (No GJB1319006). Furthermore, this research was also partially supported by Northeast Forestry University, with College Reform Project (DL

20190005). The study reflects only the authors' views and the Education Department of Heilongjiang Province and Northeast Forestry University are not liable for any use that may be made of the information contained herein.

**Acknowledgments:** Thanks to Shao-Yan Li, librarian of Northeast Forestry University, for providing helps in consulting electronic periodicals and books. Thanks to Yang Cheng, professor of the College of Engineering, University of Wisconsin-Madison, for his suggestions on the structure of the paper.

**Conflicts of Interest:** The authors declare no conflict of interest.

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
