# Peer review of "Analysis on the Higher Education Sustainability in China Based on the Comparison between Universities in China and America"

_sustainability, doi:10.3390/su12020573_

Round 1

Reviewer 1 Report

There is no applied research methods and no information about the time at which the research was conducted in the Abstract session. Supplementing this data would certainly be an encouraging element for the potential reader.

I recommend the revision and rewriting Chapter 3 Higher Education Scale in China and the United States. Please reduce the number of subchapters to 3.1, which concerns China, and the subchapter discussing issues in the USA.

In Table 10 Enrollments over the years in the USA (Million) and Figure 1 The admission of new students, I recommend limiting it to 1991-2017 (2018). The data presented in this way clearly show the trend and there is no need to reach back to a distant history.

Conclusion lacks wider insights into what this article can bring to theory and practice.

Author Response

Response to Reviewer 1 Comments

Point 1: There is no applied research methods and no information about the time at which the research was conducted in the Abstract session. Supplementing this data would certainly be an encouraging element for the potential reader.

Response 1: We added research methods and the time at which the research was conducted in the Abstract session. The modified content is “To find and solve the problems existing in the development of higher education in China, the input-output, scale of higher education, students’ tuition and teachers’ income of Chinese and American universities are compared by the methods of investigation and analysis in the last 2 years”.

Point 2: I recommend the revision and rewriting Chapter 3 Higher Education Scale in China and the United States. Please reduce the number of subchapters to 3.1, which concerns China, and the subchapter discussing issues in the USA.

Response 2: Thank you for your recommendation, and I reduced the number of subchapters to 3.1.

Point 3: In Table 10 Enrolments over the years in the USA (Million) and Figure 1 The admission of new students, I recommend limiting it to 1991-2017 (2018). The data presented in this way clearly show the trend and there is no need to reach back to a distant history.

Response 3: Table 10 has been modified according to the opinions of reviewers, and table 7 has been modified accordingly for the convenience of comparison. The modifications are shown in the latest uploaded manuscript.

Point 4: Conclusion lacks wider insights into what this article can bring to theory and practice.

Response 2: We added some content at the end of the conclusion. We believe it can make up for the above shortcomings. The added content is “Our findings are fully consistent with the actions of the Chinese government. In recent years, the Chinese government has begun to control the expansion of higher education, improve the research funds efficiency and the output of higher education, and raise the incomes of university teachers. However, as China is still a developing country, some families cannot afford to raise tuition fees significantly. Therefore, our next research goal is to help the government formulate policies that can not only raise tuition fees to maintain the sustainable development of higher education, but also guarantee the enrolment of low-income students”.

Reviewer 2 Report

I found this an interesting comparison between Chinese and American higher education, in terms of size, reputation rankings, tuition fees and salaries.  I appreciated the structure of the paper and the progression through a series of comparisons. Given the growth of the Chinese higher education system and the undoubted challenges this must pose for China, I welcome a paper that investigates these issues. I think many readers will be interested in it for its descriptive value. However, it has some substantial flaws that need to be addressed.  Throughout, I thought: 

There needs to be much more attention to differences within countries. The American system is very diverse in terms of prestige, costs of tuition, and faculty salaries, and even mission (four year undergraduate colleges versus research intensive universities). There was little recognition of this fact.  If your intent is to compare the top 10 universities of each country, that will be a much fairer comparison (apples to apples), but mixing analyses of the “average” tuition or salary in one country with the “average” tuition or salary in the other country is fraught. Or worse yet, comparing the average in China with the top 10 in the US is extremely problematic.  Although I am not as familiar with the Chinese system, it appears that there is also diversity within the Chinese sector in terms of prestige, age, public/private status and, likely, the cost of living and salaries for staff depending upon where in the country they are based (e.g. Shanghai and Beijing are likely quite different from rural parts of the country). These within-country differences are also not mentioned, but have a direct bearing on your main argument and conclusion, as laid out in the abstract.  Salaries and tuition may need to be placed in more local contexts if there are large differences in costs of living within countries and large differences between tuition and salaries at different institutions.  Although you repeatedly reference the top 10 universities, you fail to discuss the new initiatives by the Chinese government to attract world talent to the top Chinese universities with special incentives and funding to research intensive universities in China.  These new initiatives are offering salaries and packages of research support that are internationally competitive.  Is your story, then, more about the fact that there is a wealthy, prestigious set of universities in China that is getting lots of investment, but that there are large numbers of lower performing universities that have low tuition and low salaries?  If the argument is centrally about those newer, poorer universities, then it does not help your argument to compare to top universities in the US.  The US also has community colleges with low admissions standards, low tuition, and low salaries that are low prestige and are not producing research outputs.  In fact, much of the teaching at those community colleges is now undertaken by very low paid part time teachers on short term contracts, so it isn’t even a fair comparison to use the average salaries of assistant/associate/and full professors because those now represent a shrinking percentage of the people who are teaching at higher education level.  Those national averages leave out all the part time, adjunct faculty.  Maybe the system needs both ends of the spectrum when you move to massification of higher education (as China seems to have done in the past 2 decades, in particular)?     Much more attention to substantial differences between the two countries. It appears from the paper that “higher education is mainly run by the public sector, and the funding is mainly from the government at all levels.” This characteristic makes it very different from the sector in the US and suggests that a better comparison might be, for example, with, say, English HE.  English HE is also doing well in the overall university rankings, but has a standardized salary scale across the public sector (which is most of the sector) and a standardised tuition fee across subjects and universities regardless of prestige of the university.  In contrast, the American system of private higher education (into which all the top 10 universities in the US fall) is much more market driven in terms of tuition fees and salaries.  Therefore, the whole paper has the feel of comparing apples with oranges (even apples with bananas!).  The fact that these major differences in the systems are not mentioned makes the overall conclusion to mimic American higher education (with higher tuition, higher salaries) seem questionable. 

3. In addition, there are numerous specific examples of places where these kinds of nuances might be discussed or where references to the Chinese system need clarification for external readers. For example:

Table 1 – it would be better to present figures per students or per staff member to take account of size differences between the universities. Table 2 – do you mean the % of the university’s budget expenditure that comes from government? “proportion of fiscal appropriation” is ambiguous. Line 120 – gap between provincial and private. Please clarify what “provincial” means here.  Does it mean public (versus private) or is it a geographic reference (e.g. rural versus urban?). In Tables 1 and 2, do tell us which universities are private. Line 129 – leading private US colleges have large endowments through private philanthropic donations. The term “corporate sponsorship” does not describe these private donations accurately.  Thus one needs to be careful about inferring anything about government investment in higher education if you are looking at the expenditures of top private universities. Line 114 – needs to be clarified “top income sources are the same for China and US”. The income sources vary tremendously within the US higher education sector.  Are you specifically referring here to those top universities in the US, which are private?  Are you specifically referring to public or private top universities in China?   Specify which ones you are comparing here so we can be sure you are comparing like to like (apples to apples insofar as possible).  Lines 143-146 are not needed in the flow of the argument. Focus here only on the Times Higher League table data which you have in the data tables – do not stray into editorial comments, hearsay, or unsubstantiated data. Line 153 – do not say that “it is gratifying that Tsinghua” has risen in the rankings. Simply report it as a fact, do not offer an opinion of whether that is good or bad.  You might say “it is notable that…”   Lines 164-167 – there is a need to look beyond the total investments to take size into account. Also need to distinguish expenditure from government investment to make this claim. Focus on expenditure, which is what you referenced in Table 1. Lines 168-172, you rightly note different histories, although most of the leading US universities are a lot older than just 70 years old, so I don’t know where that figure came from. However, I think that the different ages or even warfare are insufficient explanations for differences between rankings. You don’t mention, for example, the dominance of English as the common language of science/academia and the way that advantages the US in terms of scientific outputs (in Anglophone/international journals which are most highly ranked) and attracting bright and privileged students and staff from around the world (which necessarily figures into the international outlook aspect of the rankings). In the discussion surrounding Table 7, can you make it explicit that expansion is associated with a higher proportion of the population attending university, rather than say, just a growing population? It appears to be a case of massification, as has been observed in many other countries (that is, larger percentages of the population attending university).  On line 199 you say that the enrolment rate is 45.7 percent, but it might be useful to make that clearer with, for example, a table of the percentage of the population of 18 year olds attending university over time or the percent of students of college age attending college. (that is, enrolment or participation rates). Lines 207-209, are you talking about the top universities in China or other universities or all universities? (Is it a heavily tiered system, as it is in the US?) Up to now you have focused on the top universities, so it would be useful to signpost when you are referring to other colleges, as we don’t see any figures about them.  Just saying “unsound management” is unclear.  What does this mean? Do you need to say that at all? As with the discussion of numbers of Chinese students in the system, it will be useful in the discussion of numbers of students in the US system to refer to enrolment rates, not absolute numbers. (although the overall scale of the system is interesting to note, it is more relevant to put it into context in relation to the population as a whole and participation rates, while also noting the number of international students). Line 249 – you declare the growth of the numbers of students in China as “a problem”. It would be more neutral to simply say it is a major “difference” between China and the US in the past two decades.  The US had a period of rapid expansion of its higher education, too, but considerably earlier than China.  In fact, it was during that expansion that community colleges grew up (with low admissions standards, low reputation, low tuition, low faculty salaries and no appreciable research activity).  You may want to include a more substantial discussion section in which you make an argument for it being a “problem”,  but that argument may be beyond the scope of this paper. You may want to consider the larger economic context, too, in which massification happened, as you refer to fewer workers available for manufacturing jobs.  Massification of higher education may be associated more generally with a shift away from manufacturing to a knowledge economy.  Perhaps if the economy is dependent upon manufacturing and is expected to continue to be so, then perhaps that is why you are one arguing against expansion of higher education in China? For a thoughtful consideration of how to use comparative higher education research to rethink national systems, see John Aubrey’s paper on the Dynamics of Massification(https://www.researchgate.net/publication/228707548_The_Dynamics_of_Massification_A_Comparative_Look_at_Higher_Education_Systems_in_the_UK_and_California)  It will also provide you with more background on the American system(s) of higher education. There seems to be a discrepancy in participation rates reported within the paper. On line 199, you report 45.7%; on line 199 it is reported as 27.19%.  Perhaps these are different statistics, but it would be useful to clarify them. Line 257 - do not interject a value judgment “growth rate is too fast”. See comment 14 above. Line 277 – again be specific that you are referring to the top 10 universities. There are much cheaper options at state universities and community colleges and tuition at also varies considerably by state.  Aubrey doesn’t try to compare all of the US with the UK, but just the California state system with the UK system. Line 247 - again, average tuition fees for those top 10 colleges in the US are not representative of all of American higher education, and it is therefore a poor comparison or benchmark to use for anything except, possibly, a private university in China that is also ranked in the top of world rankings. Line 247 – it would be useful to note, too, that there are huge scholarships given to lower income students, so that the real price tag for many students is much, much lower. It may also be worth noting the overall income inequalities in the US, which may be substantially different from China (or not?) Orient figures 2 and 3 so that the pie chart is vertical and one dimensional so it is easier to read. Lines 320 – don’t offer an opinion or judgment here with “too low”. One could equally argue that tuition fees in the US are too high.  In fact, this is a typical criticism.  Consider comparing faculty salaries with that of the “average income” within the country or to the average worker with a similar level of education in a related sector (e.g. doctors or lawyers). Line 328 – what is a “211 university” and why pick that particular university? (I see that you have provided justification on lines 373, but that should be moved up to the first time that university is mentioned. Line 341 – what is “Green Pepper”? Typo on average line of Table 16 Add caveats around lines 358 that salaries vary by discipline. In fact, noting this fact highlights that salaries are market driven and are set to attract talent, taking into account other options for people with a PhD in a particular discipline (salaries are higher in business and science than they are in the humanities or education). In contrast, salaries do not vary by discipline in the UK (except at the highest levels of the professorial scale). Do they vary by discipline in China? It is also notable that the top universities are in the most expensive urban areas of the country, with the highest costs of living, so their higher salaries may also reflect their location. Why not compare faculty salaries to average within-country salaries as you did with tuition? Consider using median income rather than mean income, given the tremendous income disparities in the US. Line 401 – “developing countries” is sufficient. Don’t refer to resource poor or low income countries as “backward”. Line 411- it seems you are referring to absolute scale, not relative scale. By relative scale (i.e. participation rates), China’s higher education is still much smaller than the US. This fact needs to be considered. Conclusion – given the concerns I have about comparability and the absence of discussion of these limitations, the conclusion seems overstated and not well-supported. It seems to be missing a discussion section in which the authors can interpret the findings. A discussion section – rather than rushing to conclusions and recommendations – would allow you to raise some of the concerns you have scattered in the findings section (“problem”, “too fast”, “too low”), while also balancing it with a more critical view of the American system, which has been criticised as unsustainable precisely for its marketization and high tuition fees.

Author Response

Response to Reviewer 1 Comments

I found this an interesting comparison between Chinese and American higher education, in terms of size, reputation rankings, tuition fees and salaries. I appreciated the structure of the paper and the progression through a series of comparisons. Given the growth of the Chinese higher education system and the undoubted challenges this must pose for China, I welcome a paper that investigates these issues. I think many readers will be interested in it for its descriptive value. However, it has some substantial flaws that need to be addressed. Throughout, I thought:

Point 1: There needs to be much more attention to differences within countries. The American system is very diverse in terms of prestige, costs of tuition, and faculty salaries, and even mission (four year undergraduate colleges versus research intensive universities). There was little recognition of this fact.

Response 1: I quite agree with you that differences within countries must be considered. Like American university system, the Chinese university system is also very diverse in terms of prestige, costs of tuition, faculty salaries, and mission. As a matter of fact, I have taken them into consideration. For example, when comparing the funding input of top 10 universities of each country, we compared the funding input in purchasing power terms. For another example, when comparing the tuition fees of Chinese and American universities, we use absolute tuition fees, relative tuition fees and tuition income ratio. There are some other examples. In the fifth chapter, the average income of Chinese university teachers was compared with that of American university teachers.

Point 2: If your intent is to compare the top 10 universities of each country, that will be a much fairer comparison (apples to apples), but mixing analyses of the “average” tuition or salary in one country with the “average” tuition or salary in the other country is fraught. Or worse yet, comparing the average in China with the top 10 in the US is extremely problematic.

Response 2: When I compare the income and tuition of college teachers, I aimed to compare the top universities with top universities, the average compare with the average. However, due to my unclear expressions, which caused your misunderstanding. For example, I listed the income of the top ten college teachers in the United States, but didn't list the income of the top ten college teachers in China, which caused you to misunderstand. I have added a lot of data and elaborated it in more detail, hoping to eliminate your misunderstanding.

Point 3: Although I am not as familiar with the Chinese system, it appears that there is also diversity within the Chinese sector in terms of prestige, age, public/private status and, likely, the cost of living and salaries for staff depending upon where in the country they are based (e.g. Shanghai and Beijing are likely quite different from rural parts of the country). These within-country differences are also not mentioned, but have a direct bearing on your main argument and conclusion, as laid out in the abstract.

Response 3: Yes, there is also diversity within the Chinese sector in many terms as you said. Therefore, at the end of section 5.1, I added the argument that the income of Chinese college teachers is also influenced by the prestige, public/private status, the cost of living in the region, the discipline and so on. Generally speaking, public universities with good social prestige, the teachers have high salary. Where the cost of living is high, college teachers are well paid. The highest-paid discipline for Chinese college teachers is banking.

Point 4: Salaries and tuition may need to be placed in more local contexts if there are large differences in costs of living within countries and large differences between tuition and salaries at different institutions.

 Response 4: The difference in income is explained above, so let's talk about tuition. Tuition fees at Chinese universities are set by the government and are only related to the major, private or public, regardless of the region. This may be a big difference from the United States. I added discussion in the last paragraph of 4.1.

Point 5: Although you repeatedly reference the top 10 universities, you fail to discuss the new initiatives by the Chinese government to attract world talent to the top Chinese universities with special incentives and funding to research intensive universities in China. These new initiatives are offering salaries and packages of research support that are internationally competitive.

Response 5: Indeed, the Chinese government has done a lot to attract talent. However, the effect is not obvious, only a few overseas Chinese return China to work. It can't attract the world's best talent like the United States. There are many reasons for this, cultural reasons, institutional reasons, and most importantly, the unimpressive level of wages in China. The number of university teachers enjoying special incentives and funding is so small that it is not discussed in the manuscript.

 Point 6: Is your story, then, more about the fact that there is a wealthy, prestigious set of universities in China that is getting lots of investment, but that there are large numbers of lower performing universities that have low tuition and low salaries?

Response 6: The reality in China is that only a few top universities get a lot of investment, while others, even key ones, do not. For example, Northeast Forestry University, a key university under the ministry of education, has an annual budget of only 5 percent of that of Tsinghua University. Another fact is that the tuition fees of both top and ordinary universities are not very different, no more than twice, and both are very low, usually only 5,000 Yuan per year. Therefore, the income gap of university teachers is huge, and most university teachers have very low income.

Point 7: If the argument is centrally about those newer, poorer universities, then it does not help your argument to compare to top universities in the US. The US also has community colleges with low admissions standards, low tuition, and low salaries that are low prestige and are not producing research outputs.  In fact, much of the teaching at those community colleges is now undertaken by very low paid part time teachers on short term contracts, so it isn’t even a fair comparison to use the average salaries of assistant/associate/and full professors because those now represent a shrinking percentage of the people who are teaching at higher education level. Those national averages leave out all the part time, adjunct faculty. Maybe the system needs both ends of the spectrum when you move to massification of higher education (as China seems to have done in the past 2 decades, in particular)?

Response 7: In fact, my focus is not just on the new, poor universities, but on the differences between Chinese and American universities. The purpose of my research is to compare Chinese and American higher education and find out the deficiency of Chinese higher education. In the process of research, we also find that there are similarities between Chinese and American higher education. Such as, Chinese and American universities are rigidly hierarchical. Chinese colleges and universities are similar to those in the United States, with distinct grades. Chinese top universities are similar to American top universities, and Chinese vocational and technical colleges are similar to American community colleges. However, the composition of college teachers in China and the United States is quite different. The United States has many part time, adjunct faculties, while China has few. The scale of higher education in China has multiplied several times in the past 20 years, realized massification of higher education. The United States also experienced the massification of higher education, but only a few decades before China.

Point 9: Much more attention to substantial differences between the two countries.

It appears from the paper that “higher education is mainly run by the public sector, and the funding is mainly from the government at all levels.” This characteristic makes it very different from the sector in the US and suggests that a better comparison might be, for example, with, say, English HE.  English HE is also doing well in the overall university rankings, but has a standardized salary scale across the public sector (which is most of the sector) and a standardised tuition fee across subjects and universities regardless of prestige of the university. In contrast, the American system of private higher education (into which all the top 10 universities in the US fall) is much more market driven in terms of tuition fees and salaries. Therefore, the whole paper has the feel of comparing apples with oranges (even apples with bananas!). The fact that these major differences in the systems are not mentioned makes the overall conclusion to mimic American higher education (with higher tuition, higher salaries) seem questionable.

Response 9: You're right. Higher education in China is mainly run by the public sector. For example, the top 200 universities are all public. By contrast, the most top 200 American universities are private. The funding in China is also mainly from the government at all levels. These are very different from American universities. China is a developing country with limited financial resources, and higher education relying mainly on government support is not sustainable. Compared with the United States, Chinese college teachers' incomes and students' tuition are very low. These are extremely detrimental to the sustainable development of higher education in China. The situation in China is different from that of the United States, and we are not suggesting that Chinese university tuition be raised to the same level as in the United States. We only hope to raise teachers' income and students' tuition fees step by step on the basis of full investigation and research, so as to alleviate the shortage of development funds for non-top universities and promote the sustainable development of higher education in China. On the whole, higher education in the United States has been a success, attracting talented people from all over the world and promoting the development of American economy and technology. To make the comparing more appropriately(apples to apples), a lot of data and arguments were added to the manuscript.

Point 10: In addition, there are numerous specific examples of places where these kinds of nuances might be discussed or where references to the Chinese system need clarification for external readers. For example: Table 1 – it would be better to present figures per students or per staff member to take account of size differences between the universities.

Response 10: Thank you very much for your advice. If we consider the number of students or employees in Chinese universities, we should consider the number of students or employees in American universities. However, I have returned to China from the United States, and it is difficult for me to obtain the data of students or employees in American universities due to the Internet block. Therefore, I am sorry to say that I have to consider only the total budget.

Point 11: Table 2 – do you mean the % of the university’s budget expenditure that comes from government? “proportion of fiscal appropriation” is ambiguous.

Response 11: Yes, the content of Table 2 refers to the proportion of financial allocation in total expenditure. To reduce ambiguity, “Proportion of fiscal appropriation” has been changed to “Proportion of fiscal appropriation in total expenditure”.

Point 12: Line 120 – gap between provincial and private. Please clarify what “provincial” means here.  Does it mean public (versus private) or is it a geographic reference (e.g. rural versus urban?).

Response 12: Provincial colleges and universities mean that the provincial government where the colleges and universities are located provides financial support. Provincial colleges and universities, like those under the ministry of education ones, are also public colleges and universities. In China, the best colleges and universities are under the ministry of education, then provincial ones, and the worst are private colleges and universities.

 Point 13:In Tables 1 and 2, do tell us which universities are private.

Response 13: In China, the top 200 universities are all public universities. Therefore, both table 1 and table 2 are public universities.

Point 14: The term “corporate sponsorship” does not describe these private donations accurately. Thus one needs to be careful about inferring anything about government investment in higher education if you are looking at the expenditures of top private universities.

Response 3.2: “corporate sponsorship” has been changed to “private donations”.

Point 15: Line 114 – needs to be clarified “top income sources are the same for China and US”. The income sources vary tremendously within the US higher education sector.  Are you specifically referring here to those top universities in the US, which are private?  Are you specifically referring to public or private top universities in China?   Specify which ones you are comparing here so we can be sure you are comparing like to like (apples to apples insofar as possible). 

Response 15: Yes, I am referring to those top universities in the US and China. It doesn't matter whether it's public or private. In fact, the top universities in the United States are private, while the top universities in China are public.

Point 16: Lines 143-146 are not needed in the flow of the argument. Focus here only on the Times Higher League table data which you have in the data tables – do not stray into editorial comments, hearsay, or unsubstantiated data.

Response 16: I have deleted this part.

Point 17: Line 153 – do not say that “it is gratifying that Tsinghua” has risen in the rankings. Simply report it as a fact, do not offer an opinion of whether that is good or bad.  You might say “it is notable that…”  

Response 17: Thank you very much for your advice. “it is gratifying that Tsinghua” has been changed to “it is notable that…”.

Point 18: Lines 168-172, you rightly note different histories, although most of the leading US universities are a lot older than just 70 years old, so I don’t know where that figure came from. However, I think that the different ages or even warfare are insufficient explanations for differences between rankings.

Response 18: Yes, most of the leading US universities are a lot older than just 70 years old, some of them up to 200 years and more. Harvard University, College of William & Mary, St. John's College and Yale University were founded over 300 years ago. According to statistics, there are about 100 colleges and universities in the United States with a history of more than 100 years. But we referred to the average age of 4724 colleges and universities in the United States, many of them are less than 70 years old. In my opinion, the history of the university and the war has something to do with the development level of the university. The long history of a university means that it has experienced a long period of development, and the war is very destructive to the development of universities.

Point 19: You don’t mention, for example, the dominance of English as the common language of science/academia and the way that advantages the US in terms of scientific outputs (in Anglophone/international journals which are most highly ranked) and attracting bright and privileged students and staff from around the world (which necessarily figures into the international outlook aspect of the rankings).

Response 19: Yes, the ranking needs to consider the language factor. Universities are ranked on the basis of teaching effectiveness, research results and international impact, most of which are presented in English, giving British and American universities strong advantages in ranking. On the contrary, universities in Germany, Japan, China and other non-English-speaking countries are at a disadvantage in ranking. Generally speaking, although the ranking is influenced by language, it still has certain reference value. I have added this argument to the manuscript.

Point 20: In the discussion surrounding Table 7, can you make it explicit that expansion is associated with a higher proportion of the population attending university, rather than say, just a growing population? It appears to be a case of massification, as has been observed in many other countries (that is, larger percentages of the population attending university). 

Response 20: Yes, I had made it explicit that expansion is associated with a higher proportion of the population attending university. Since the new students’ admission in 2018 is almost 13 times that of 1989. In comparison, from 1989 to 2018, China's population increased from 1.12 billion to 1.39 billion, only increased by 24.11 %. In the past 30 years, the massification of higher education in China has developed rapidly, as has been happened in many other countries.

Point 21: On line 199 you say that the enrolment rate is 45.7 percent, but it might be useful to make that clearer with, for example, a table of the percentage of the population of 18 year olds attending university over time or the percent of students of college age attending college. (that is, enrolment or participation rates).

Response 21: To make it clearer that the enrolment rate here means the percent of students of college age attending college, I have added a table (Table 8) to list the percent of students of college age attending college.

Point 22:Lines 207-209, are you talking about the top universities in China or other universities or all universities? (Is it a heavily tiered system, as it is in the US?) Up to now you have focused on the top universities, so it would be useful to signpost when you are referring to other colleges, as we don’t see any figures about them.  Just saying “unsound management” is unclear.  What does this mean? Do you need to say that at all? As with the discussion of numbers of Chinese students in the system, it will be useful in the discussion of numbers of students in the US system to refer to enrolment rates, not absolute numbers. (although the overall scale of the system is interesting to note, it is more relevant to put it into context in relation to the population as a whole and participation rates, while also noting the number of international students).

Response 22: In this part, we are talking about all universities. This paper discusses the problems in higher education in China, not just in top universities. Problems mentioned here exist in most colleges and universities, especially in provincial colleges and vocational colleges. The paper mentions the top universities many times because the top universities are the representatives of Chinese universities. I just take them as an example. In this part, we mentioned the 10 largest universities with the most students. What I am saying here is that there are many large universities in China, which cause problems of insufficient supply of resources and chaotic management. It has nothing to do with enrolment rates here. The number of foreign students studying in the United States is more than one million, while in China it is less than 50,000. This is very different from America, so there is no need to introduce the number of foreign students.

Point 23:Line 249 – you declare the growth of the numbers of students in China as “a problem”. It would be more neutral to simply say it is a major “difference” between China and the US in the past two decades.  The US had a period of rapid expansion of its higher education, too, but considerably earlier than China.  In fact, it was during that expansion that community colleges grew up (with low admissions standards, low reputation, low tuition, low faculty salaries and no appreciable research activity).  You may want to include a more substantial discussion section in which you make an argument for it being a “problem”, but that argument may be beyond the scope of this paper. You may want to consider the larger economic context, too, in which massification happened, as you refer to fewer workers available for manufacturing jobs.  Massification of higher education may be associated more generally with a shift away from manufacturing to a knowledge economy.  Perhaps if the economy is dependent upon manufacturing and is expected to continue to be so, then perhaps that is why you are one arguing against expansion of higher education in China? For a thoughtful consideration of how to use comparative higher education research to rethink national systems, see John Aubrey’s paper on the Dynamics of Massification(https://www.researchgate.net/publication/228707548_The_Dynamics_of_Massification_A_Comparative_Look_at_Higher_Education_Systems_in_the_UK_and_California)  It will also provide you with more background on the American system(s) of higher education.

Response 23: Thank you very much for recommending a good article. I read the paper carefully and found that the popularization of higher education in the United States took place in the 1960s and 1970s. There are some similarities between the expansion of Chinese and American universities, but China's problems are far more serious. After the expansion, there are nearly 2700 universities in China, but more than 2,000 are vocational and technical colleges and only 600 undergraduate schools. After the expansion, community colleges had grow up in the United States, they have low admission standards, low salary of teachers and little scientific achievements, which are similar to vocational and technical colleges in China. However, the difference is that the tuition fees of vocational and technical colleges in China are higher than those of undergraduate universities, which are generally about twice as high as those of undergraduate universities. In addition, the staffs in vocational and technical colleges in China are all full-time, and the government needs to pay full salaries, so the expansion of universities adds a great burden to the government. I am not opposed to the expansion of universities in China, but I am concerned about the problems caused by the rapid expansion of universities. We are soberly aware that China still needs to rely on manufacturing for a long time, and the knowledge economy is still far behind that of the United States. The labour shortage caused by the expansion of Chinese universities has taught the Chinese government a lesson, with manufacturing costs rising and many manufacturing plants already moving to Southeast Asia. The expansion of Chinese universities has reduced GDP growth from a peak of 15 % to about 6 % today.

Point 24:There seems to be a discrepancy in participation rates reported within the paper. On line 199, you report 45.7%; on line 199(line 243,not 199) it is reported as 27.19%.  Perhaps these are different statistics, but it would be useful to clarify them.

Response 24: Thank you very much for pointing out the mistake. On line 243, it should be “accounting for 27.19‰ of the total population”.  It is persiflage, not percent, and it is total population rather than student-age population. I've corrected in the manuscript.

Point 25: Line 257 - do not interject a value judgment “growth rate is too fast”. See comment 14 above.

Response 25:I agree with you, it should not be interject value judgment. I've corrected it.

Point 26: Line 277 – again be specific that you are referring to the top 10 universities. There are much cheaper options at state universities and community colleges and tuition at also varies considerably by state.  Aubrey doesn’t try to compare all of the US with the UK, but just the California state system with the UK system. Line 247 - again, average tuition fees for those top 10 colleges in the US are not representative of all of American higher education, and it is therefore a poor comparison or benchmark to use for anything except, possibly, a private university in China that is also ranked in the top of world rankings.

Response 26:I am very sorry that I overlooked the differences in tuition fees between different universities in the United States. Most of America's top universities are private, paying two to three times as much as state or community colleges. The top universities in China are public universities, which often charge half as much as vocational and technical colleges. In the manuscript, I compared not only the average tuition of the top universities in China and the United States, but also the average tuition of universities in China and the United States.

Point 27:Line 247 – it would be useful to note, too, that there are huge scholarships given to lower income students, so that the real price tag for many students is much, much lower. It may also be worth noting the overall income inequalities in the US, which may be substantially different from China (or not?)。

 Response 27:Here we only compare the tuition, of course, scholarships can also be part of the tuition, so that the real burden of American students is not so heavy. China also has a huge wealth gap, but because tuition fees are so low, most families can afford it and few take out loans to go to college.

Point 28: Orient figures 2 and 3 so that the pie chart is vertical and one dimensional so it is easier to read.

Response 28:Figure 2 and figure 3 have been redrawn.

Point 29: Lines 320 – don’t offer an opinion or judgment here with “too low”. One could equally argue that tuition fees in the US are too high.  In fact, this is a typical criticism.  Consider comparing faculty salaries with that of the “average income” within the country or to the average worker with a similar level of education in a related sector (e.g. doctors or lawyers).

Response 29:I am very sorry that I have made the same mistake many times, I shouldn't be critical. This statement has been modified. To make the salaries of university teachers comparable, I also added the revenue ranking of major industries in China.

Point 30: Line 328 – what is a “211 university” and why pick that particular university? (I see that you have provided justification on lines 373, but that should be moved up to the first time that university is mentioned.

Response 30:“211 project” refers to build about 100 key universities in the 21st century in China. I explained that in the manuscript.

Point 31: Line 341 – what is “Green Pepper”?

Response 31:“Green Pepper” refers to the young college teachers, whose education is high, income is low, and is full of high work pressure.

Point 32:Typo on average line of Table 16。

Response 32:Yes, there was an obvious mistake, I have corrected it.

Point 33: Add caveats around lines 358 that salaries vary by discipline. In fact, noting this fact highlights that salaries are market driven and are set to attract talent, taking into account other options for people with a PhD in a particular discipline (salaries are higher in business and science than they are in the humanities or education). In contrast, salaries do not vary by discipline in the UK (except at the highest levels of the professorial scale). Do they vary by discipline in China? It is also notable that the top universities are in the most expensive urban areas of the country, with the highest costs of living, so their higher salaries may also reflect their location.

Response 33:I have added caveats that salaries of teachers in American colleges and universities depend not only on their titles and the schools they work for, but also depend on their discipline and the region in which they work. And I add list the incomes of university teachers of different disciplines in the United States. There is also a large gap in the pay of university teachers in China, which is influenced by almost the same factors as in the United States. I am surprised that the pay of British university teachers is not affected by their subjects.

Point 34: Why not compare faculty salaries to average within-country salaries as you did with tuition? Consider using median income rather than mean income, given the tremendous income disparities in the US.

Response 34:Thank you for your good suggestion. I supplemented and compared the revenue of the major domestic industries and made a brief comment.

Point 35: Line 401 – “developing countries” is sufficient. Don’t refer to resource poor or low income countries as “backward”.

Response 35:I'm sorry, I respect all countries, but my statement is misleading. I have revised it.

Point 36: Line 411- it seems you are referring to absolute scale, not relative scale. By relative scale (i.e. participation rates), China’s higher education is still much smaller than the US. This fact needs to be considered.

Response 36:I am referring to absolute scale here, not relative scale, I have revised it.

Point 37:Conclusion – given the concerns I have about comparability and the absence of discussion of these limitations, the conclusion seems overstated and not well-supported. It seems to be missing a discussion section in which the authors can interpret the findings. A discussion section – rather than rushing to conclusions and recommendations – would allow you to raise some of the concerns you have scattered in the findings section (“problem”, “too fast”, “too low”), while also balancing it with a more critical view of the American system, which has been criticised as unsustainable precisely for its marketization and high tuition fees. 

Response 37:Some data and discuss are added to the manuscript, which is helpful to support the conclusion. Some of the statements in the manuscript were inappropriate and I have corrected them. For example, “growth rate is too fast” has been changed to “growth rate is far more than demand”. “too low” has been changed to “ to be improved”. The development model of American universities has some shortcomings, but it is generally successful and has certain reference significance for developing countries.

Round 2

Reviewer 2 Report

REVIEWER 1 Response to Revision

The additions in sections 3.1, 3.2, 4.1. 4.2 and 4.3 are substantial improvements and go a long way toward addressing my concerns.  However, for many of my comments, you have responded to me, but have not added those further explanations into the papers. Those need to be done.  For example:

Line 20 – the new section in red confuses rather than clarifies.  It is not needed.

Line 24 – “is to be improved” is ambiguous.  What constitutes “improvement”?  If you think it needs to be raised, say “increased”.

Line 24- absolute tuition is only 5.3% - a summary that references relative tuition would be more useful.

Line 43 – you have usefully defined provincial in your response to my point 12.  This definition/explanation also needs to be added to the paper for other readers who may be unclear on this point.

Line 45 – “non-key university” also needs to be defined/explained here.  Again, you have provided considerable clarification about the Chinese system to me in your response letter which other readers also need.  Do not assume that your readers are familiar with the Chinese system.  Take my questions as indicative of the kinds of questions other readers will have and which your paper needs to explain.

line 107 – “double first class universities” – again, this is unclear to a reader unfamiliar with the Chinese system.  Again, a short description of the context of Chinese higher education would help clarify what you are researching before you get into your charts and comparisons.

Line 134 – another place you could clarify that the top Chinese universities are public.

Lines 110-115.  MIT had 11,376 students in 2018 compared to Tsinghua’s 48,739 registered students.  Therefore, if they are spending similar amounts, Tsinghua’s investment per student is one quarter that of MIT’s, not the same.  Numbers of students, according to web research is:  Yale=12,385 (2015); Harvard=22,947 (2017); Princeton=8,623 (2017); Stanford=16,924 (2017); University of Washington=46,081 (2016); Caltech=2,238 (2017).   Zhejiang University = 54,641 Shanghai University=56,753 Jiaotong University = 42,881(2011).  Given that the leading American private universities you are comparing expenditures to are much smaller than the leading Chinese universities, your claims that investments are similar is misleading.  These claims are made in the abstract and do not seem to be borne out when you compare the relative size of the institutions. These figures underscore my point that it is not appropriate to compare leading Chinese public universities with leading US private universities.  The best comparison in this collection you are making is with the University of Washington, which also happens to be a public university and is ranked 22nd in the US News rankings of public universities.  Other leading public universities are more similar in size to the leading Chinese universities. According to the 2020 USNews.Com rankings of top US public universities:  1) University of California – Los Angeles =31,577 students; 2) University of California Berkeley= 30, 853; 3) University of Michigan- Ann Arbor = 30, 318; 4) University of Virginia=16,777. 5) Georgia Institute of Technology=16,049. 

Line 116 – is insufficiently contextualised in claiming similar income sources.  See my original, overarching Points 1 and 9, which are not adequately addressed in the whole of section  2.1.   All of the claims made in section 2.3 rest upon the assumption that Chinese universities are funded to the same levels as the top US private universities, but Section 2.1 has not shown that sufficiently.

Lines 150-151 – delete the comment “Generally speaking, 151 although the ranking is influenced by language, It's a little unfair,it still has certain reference value.”  The point is made better in the previous sentences and this new sentence is not well formed.

Line 307-308  – again you make the comparison between the top 10 universities in the US and incorrectly conclude that: “It means that the tuition income ratio of Chinese universities is only 20.21% that of the United States.” This in not compelling and distracts from your argument. Instead, compare to the average tuition you have added at line 297.

Line 339 – Table 15 showing how university teachers rank against other professions in China is useful to put your salary arguments into context.  However, without comparable data from the US, it is not doing a lot to advance your argument.  For instance, I looked at Investopedia at the top 25 professions in the US for income and, even allowing for the fact that 13 of those professions were doctors, university professors are not listed.  That is, they are not in the remaining 12 in the top 25 highest paid professions.  So, the fact that university teachers rank 10th for salary in China doesn’t suggest that they are massively underpaid relative to other professions.  Yes, I agree that it seems wrong that other professions with lower levels of education get paid more, but that happens in the US (and elsewhere), too. You seem to be saying that you can learn lessons from the US – if so, the lesson is that university professors also make less than bankers, doctors, IT managers, and so on in the US. It certainly adds rigour to your argument, but the facts don’t actually support your arguments and conclusions very well. (People pursue university teaching out of love, not because it is highly lucrative.)

Lines 428-442 it seems that, given the gap between average salaries in China and average salaries in the US (20%) that the difference in faculty salaries between the two countries is worse than the gap between the average worker in China and the average American worker, but not by a tremendous amount (Chinese faculty members make 16-20% of American faculty members; the average worker in China makes 29% of the average worker in the US.)   This analysis needs to be reflected in your conclusion on lines 474-476.  You do have some evidence to suggest Chinese faculty members need higher salaries to match their American counterparts, but not as substantially as you are suggesting in the conclusion.  The conclusion needs to be matched to your new data.

Lines 462-466 are based on the dubious sections 2.1 and 2.3. These lines still need to be amended.

Lines 469-471 do not seem to follow from the analyses presented.  You show that there is increasing participation in higher education, which suggests increasing demand for a higher education by young people and their families.  I think you are actually concerned that there is not a demand for higher education graduates in the labour force, but you have not provided any analyses of this issue, so it seems inappropriate to conclude with this as a key point. 

After a careful read of the entire manuscript, the most relevant and convincing argument you have put forward is the need to raise tuition fees.  There may also be a case for raising faculty salaries, but the data you present is less convincing here and your conclusions are overstated.  I remain unconvinced that raising tuition and raising salaries will overcome productivity gaps, as you seem to suggest by the inclusion of section 2. (Perhaps Section 2 could be removed). You have offered some very useful discussion of the problem of rapid expansion in your responses to me, but these have not been incorporated into the discussion, nor have you returned to your literature review in writing your discussion or reaching your conclusions.  Therefore, my point 37 remains unaddressed. As this is the heart of the argument of the paper – which is also reproduced in the abstract – I cannot recommend acceptance in its current form.

Author Response

Response to Reviewer 1 Comments

We are very lucky to meet such a conscientious and responsible reviewer. Thank you very much for your comments and Suggestions. We have revised all the contents according to your requirements. I hope the revised content will satisfy you. If there are still problems to be revised, please give me another chance. We will try our best to revise the paper.

Point 1: The additions in sections 3.1, 3.2, 4.1. 4.2 and 4.3 are substantial improvements and go a long way toward addressing my concerns.  However, for many of my comments, you have responded to me, but have not added those further explanations into the papers. Those need to be done.  For example:

Line 20 – the new section in red confuses rather than clarifies.  It is not needed.

Response 1: Thank you for your suggestion. We have deleted this part.

Point 2:  Line 24 – “is to be improved” is ambiguous.  What constitutes “improvement”?  If you think it needs to be raised, say “increased”.

Response 2: Yes, it is ambiguous, and we have revised.

Point 3: Line 24- absolute tuition is only 5.3% - a summary that references relative tuition would be more useful.

Response 3: Yes, a summary that references relative tuition would be more useful. So we added the relative tuition summary. 

Point 4: Line 43 – you have usefully defined provincial in your response to my point 12.  This definition/explanation also needs to be added to the paper for other readers who may be unclear on this point.

Response 4: The explanation of provincial colleges and universities has been added to the paper.

Point 5: Line 45 – “non-key university” also needs to be defined/explained here.  Again, you have provided considerable clarification about the Chinese system to me in your response letter which other readers also need.  Do not assume that your readers are familiar with the Chinese system.  Take my questions as indicative of the kinds of questions other readers will have and which your paper needs to explain.

Response 5: In china, “non-key university” mainly includes less-developed provincial and private colleges and universities. So we substituted “less-developed provincial and private colleges and universities” for “non-key university”.

Point 6: line 107 – “double first class universities” – again, this is unclear to a reader unfamiliar with the Chinese system.  Again, a short description of the context of Chinese higher education would help clarify what you are researching before you get into your charts and comparisons.

Response 6: The Chinese government has implemented a project to build first class universities and first class disciplines. The relevant universities are double first-class universities. The explanation of “double first class universities” has been added to the paper.

Point 7: Line 134 – another place you could clarify that the top Chinese universities are public.

Response 7: At the end of the paragraph, I add “In contrast, the top 20 most popular Chinese universities are all public”.

Point 8: Lines 110-115.  MIT had 11,376 students in 2018 compared to Tsinghua’s 48,739 registered students.  Therefore, if they are spending similar amounts, Tsinghua’s investment per student is one quarter that of MIT’s, not the same.  Numbers of students, according to web research is:  Yale=12,385 (2015); Harvard=22,947 (2017); Princeton=8,623 (2017); Stanford=16,924 (2017); University of Washington=46,081 (2016); Caltech=2,238 (2017).   Zhejiang University = 54,641 Shanghai University=56,753 Jiaotong University = 42,881(2011).  Given that the leading American private universities you are comparing expenditures to are much smaller than the leading Chinese universities, your claims that investments are similar is misleading.  These claims are made in the abstract and do not seem to be borne out when you compare the relative size of the institutions. These figures underscore my point that it is not appropriate to compare leading Chinese public universities with leading US private universities.  The best comparison in this collection you are making is with the University of Washington, which also happens to be a public university and is ranked 22nd in the US News rankings of public universities.  Other leading public universities are more similar in size to the leading Chinese universities. According to the 2020 USNews.Com rankings of top US public universities:  1) University of California – Los Angeles =31,577 students; 2) University of California Berkeley= 30, 853; 3) University of Michigan- Ann Arbor = 30, 318; 4) University of Virginia=16,777. 5) Georgia Institute of Technology=16,049. 

Response 8: Thank you very much for providing so much data. To compare the average budget for each student, we also needed the total budget of these colleges and universities, which we haven’t found. However, we found the total number of students and the total budget of other colleges, and then we made a comparison in Table 2.

Point 9: Line 116 – is insufficiently contextualised in claiming similar income sources.  See my original, overarching Points 1 and 9, which are not adequately addressed in the whole of section 2.1.   All of the claims made in section 2.3 rest upon the assumption that Chinese universities are funded to the same levels as the top US private universities, but Section 2.1 has not shown that sufficiently.

Response 9: We read the Points 1 and 9 of your original opinion carefully, and added a background about the income sources. The last version of the manuscript stated this part incorrectly. As a matter of fact, the total budgets of Chinese and American universities are similar, but the average student budget of China's top universities is much lower than that of American universities. Correspondingly, contents of 2.3 have also been modified.

Point 10: Lines 150-151 – delete the comment “Generally speaking, 151 although the ranking is influenced by language, It's a little unfair, it still has certain reference value.”  The point is made better in the previous sentences and this new sentence is not well formed.

 Response 10:  We have deleted this sentence.

Point 11: Line 307-308  – again you make the comparison between the top 10 universities in the US and incorrectly conclude that: “It means that the tuition income ratio of Chinese universities is only 20.21% that of the United States.” This in not compelling and distracts from your argument. Instead, compare to the average tuition you have added at line 297.

 Response 11: I'm sorry we made a mistake here. In the last paragraph of 4.1, we already mentioned “the average tuition accounts for only 12.09% that of the average annual income”. In the last paragraph of 4.2, we also mentioned average tuition fees accounts for up to 59.83% of average annual income. So, in 4.3, it should be average tuition income ratio of all colleges and universities, not top 10 universities. The result, the tuition income ratio of Chinese universities is only 20.21% that of the United States is correct.

Point 12: Line 339 – Table 15 showing how university teachers rank against other professions in China is useful to put your salary arguments into context.  However, without comparable data from the US, it is not doing a lot to advance your argument.  For instance, I looked at Investopedia at the top 25 professions in the US for income and, even allowing for the fact that 13 of those professions were doctors, university professors are not listed.  That is, they are not in the remaining 12 in the top 25 highest paid professions.  So, the fact that university teachers rank 10th for salary in China doesn’t suggest that they are massively underpaid relative to other professions.  Yes, I agree that it seems wrong that other professions with lower levels of education get paid more, but that happens in the US (and elsewhere), too. You seem to be saying that you can learn lessons from the US – if so, the lesson is that university professors also make less than bankers, doctors, IT managers, and so on in the US. It certainly adds rigour to your argument, but the facts don’t actually support your arguments and conclusions very well. (People pursue university teaching out of love, not because it is highly lucrative.)

 Response 12: In 5.1, we talk about the income of college teachers In China. So we compared the incomes of different industries in China. We don't think it's appropriate to talk about the income of American industries here. As you said, American college teachers are not well paid either. And it can't support my argument, so we thought it would be better not to talk about American industry income.

Point 13: Lines 428-442 it seems that, given the gap between average salaries in China and average salaries in the US (20%) that the difference in faculty salaries between the two countries is worse than the gap between the average worker in China and the average American worker, but not by a tremendous amount (Chinese faculty members make 16-20% of American faculty members; the average worker in China makes 29% of the average worker in the US.)   This analysis needs to be reflected in your conclusion on lines 474-476.  You do have some evidence to suggest Chinese faculty members need higher salaries to match their American counterparts, but not as substantially as you are suggesting in the conclusion.  The conclusion needs to be matched to your new data.

Response 13: Your suggestion makes a lot of sense, so I added the following sentence. National average income in China is 29.31 % of the average income in the United States. The average income of Chinese college teachers’ percentage of American college teachers is much less than the average income of Chinese national percentage of the United States.

Point 14: Lines 462-466 are based on the dubious sections 2.1 and 2.3. These lines still need to be amended.

Response 14: Yes, we have amended these lines.

Point 15: Lines 469-471 do not seem to follow from the analyses presented.  You show that there is increasing participation in higher education, which suggests increasing demand for a higher education by young people and their families.  I think you are actually concerned that there is not a demand for higher education graduates in the labour force, but you have not provided any analyses of this issue, so it seems inappropriate to conclude with this as a key point.  

Response 15: In the last part of 3.1, we added the comparison of China's graduate unemployment rate with the national unemployment rate, and the income of college graduates with that of construction workers and factory workers. I think these comparisons are helpful in reaching this conclusion.

Point 16: After a careful read of the entire manuscript, the most relevant and convincing argument you have put forward is the need to raise tuition fees.  There may also be a case for raising faculty salaries, but the data you present is less convincing here and your conclusions are overstated.  I remain unconvinced that raising tuition and raising salaries will overcome productivity gaps, as you seem to suggest by the inclusion of section 2. (Perhaps Section 2 could be removed). You have offered some very useful discussion of the problem of rapid expansion in your responses to me, but these have not been incorporated into the discussion, nor have you returned to your literature review in writing your discussion or reaching your conclusions.  Therefore, my point 37 remains unaddressed. As this is the heart of the argument of the paper – which is also reproduced in the abstract – I cannot recommend acceptance in its current form.

Response 16:  Yes, raising tuition is one of our points of this paper. By raising tuition fees, we can increase the income of teachers, attract more talents to work in universities, and ensure the sustainable development of Chinese universities. To support my point, we added a lot of data and analysis comments. In addition, we have also included the discussion on the rapid expansion of colleges and universities in the paper.I have read point 37 carefully, and we agree with you very much. America's higher education system is flawed. But there's no denying, the United States has the largest number of top universities in the world. At present, there are still many aspects worthy of reference for developing countries, especially for China.

Round 3

Reviewer 2 Report

Most of my major concerns have been addressed in this revision. 

The entire paper still needs a final editing for clarity in use of the English language, though.  New text that has been added through the revision process is particularly in need of attention from a native English speaker. As just one example:  lines 518-520, the sentence is not well-formed, which makes it very confusing: 

"However, the difference is that the tuition fees of vocational and technical
colleges in China are higher than those of undergraduate universities, which are generally about twice as high as those of undergraduate universities."

Do you simply mean, "However, unlike the US, the tuition fees of vocational and technical universities are about twice as high as those of undergraduate universities."  ?

Author Response

Response to Reviewer 2 Comments

Point 1: The entire paper still needs a final editing for clarity in use of the English language, though.  New text that has been added through the revision process is particularly in need of attention from a native English speaker. As just one example:  lines 518-520, the sentence is not well-formed, which makes it very confusing: 

"However, the difference is that the tuition fees of vocational and technical
colleges in China are higher than those of undergraduate universities, which are generally about twice as high as those of undergraduate universities."

Do you simply mean, "However, unlike the US, the tuition fees of vocational and technical universities are about twice as high as those of undergraduate universities."  ?

Response 1: Thank you very much for your suggestion to polish the manuscript again. I have entrusted my American friend Mr Ben to help me revise and polish the whole text. Since there are many revisions, please refer to the manuscript for the revised part. I hope the revised content will satisfy you. If there are still problems to be revised, please give me another chance. We will try our best to revise the paper.
